**Carbon Monitoring System Flux Net Biosphere Exchange 2020 (CMS-Flux NBE 2020)**

Junjie Liu[1,2*], Latha Baskaran[1], Kevin Bowman[1], David S. Schimel[1], A. Anthony Bloom[1], Nicholas C. Parazoo[1], Tomohiro Oda[3,4], Dustin Carroll[5], Dimitris Menemenlis[1], Joanna Joiner[6], Roisin Commane[7], Bruce Daube[8], Lucianna V. Gatti[9], Kathryn McKain[10,11], John Miller[10], Britton B. Stephens[12], Colm Sweeney[10], Steven Wofsy[8],

1. Jet Propulsion Laboratory, Caltech, CA
2. Caltech, CA
3. Global Modeling and Assimilation Office, NASA Goddard Space Flight Center
4. Goddard Earth Sciences Technology and Research, Universities Space Research Association, Columbia, MD
5. Moss Landing Marine Laboratories, San José State University, California, CA
6. Laboratory for Atmospheric Chemistry and Dynamics, NASA Goddard Space Flight Center
7. Lamont-Doherty Earth Observatory of Columbia University, NY
8. Harvard University, Cambridge, MA
9. LaGEE, CCST, INPE- National Institute for Space Research, Brazil
10. NOAA, Global Monitoring Laboratory, Boulder, CO 80305
11. University of Colorado, Cooperative Institute for Research in Environmental Sciences, Boulder, CO
11. National Center for Atmospheric Research, Boulder, CO  80301

Correspondence: Junjie Liu (junjie.liu@jpl.nasa.gov)

**Abstract**. Here we present a global and regionally-resolved terrestrial net biosphere exchange (NBE) dataset with corresponding uncertainties between 2010–2018: CMS-Flux NBE 2020. It is estimated using the NASA Carbon Monitoring System Flux (CMS-Flux) top-down flux inversion system that assimilates column $CO_2$ observations from the Greenhouse gases Observing SATellite (GOSAT) and NASA's Observing Carbon Observatory -2 (OCO-2). The regional monthly fluxes are readily accessible as tabular files, and the gridded fluxes are available in NetCDF format. The fluxes and their uncertainties are evaluated by extensively comparing the posterior $CO_2$ mole fractions with $CO_2$ observations from aircraft and the NOAA marine boundary layer reference sites. We describe the characteristics of the dataset as global total, regional climatological mean, and regional annual fluxes and seasonal cycles. We find that the global total fluxes of the dataset agree with atmospheric $CO_2$ growth observed by the surface-observation network within uncertainty. Averaged between 2010 and 2018, the tropical regions range from close-to neutral in tropical South America to a net source in Africa; these contrast with the extra-tropics, which are a net sink of $2.5 \pm 0.3$ gigaton carbon per year. The regional satellite-constrained NBE estimates provide a unique perspective for understanding the terrestrial biosphere carbon dynamics and monitoring changes in regional contributions to the changes of atmospheric $CO_2$ growth rate. The gridded and regional aggregated dataset can be accessed at: https://doi.org/10.25966/4v02-c391 (Liu et al., 2020).

## 1 Introduction

New "top-down" inversion frameworks that harness satellite observations provide an important complement to global aggregated fluxes (e.g., Global Carbon Budget (GCB), Friedlingstein et al., 2019) and inversions based on surface $CO_2$ observations (e.g., Chevallier et al., 2010), especially over the tropics and the Southern Hemisphere (SH) where conventional surface $CO_2$ observations are sparse. The net biosphere exchange (NBE), which is the net carbon flux of all the land-atmosphere exchange processes except fossil fuel emissions, is far more variable and uncertainty than ocean fluxes (Lovenduski and Bonan, 2017) or fossil fuel emissions (Yin et al, 2019), and is thus the focus of this dataset estimated from a top-down atmospheric $CO_2$ inversion of satellite column $CO_2$ dry-air mole fraction ($X_{CO2}$). Here, we present the global and regional NBE as a series of maps, time series and tables, and disseminate it as a public dataset for further analysis and comparison to other sources of flux information. The gridded NBE dataset and its uncertainty, air-sea fluxes, and fossil fuel emissions are also available, so that users can calculate carbon budget from regional to global scale. Finally, we provide a comprehensive evaluation of both mean and uncertainty estimates against the $CO_2$ observations from independent airborne datasets and the NOAA marine boundary layer (MBL) reference sites (Conway et al., 1994).

Global top-down atmospheric $CO_2$ flux inversions have been historically used to estimate regional terrestrial NBE. They make uses of the spatiotemporal variability of atmospheric $CO_2$, which is dominated by NBE, to infer net carbon exchange at the surface (Chevallier et al., 2005; Baker et al., 2006; Liu et al., 2014). The accuracy of the NBE from top-down flux inversions is determined by the density and accuracy of the $CO_2$ observations, the accuracy of modeled atmospheric transport, and knowledge of the prior uncertainties of the flux inventories.


For $CO_2$ flux inversions based on high precision *in situ* and flask observations, the measurement
error is low (<0.2 parts per million (ppm)) and not a significant source of error; however, these
observations are limited spatially, and are concentrating primarily over North America (NA) and
Europe (Crowell et al., 2019). Satellite $X_{CO2}$ from $CO_2$-dedicated satellites, such as the Greenhouse
Gases Observing Satellite (GOSAT) (launched in July 2009) and the Observing Carbon
Observatory 2 (OCO-2) (Crisp et al., 2017) have much broader spatial coverage (O'Dell et al.,
2018), which fill the observational gaps of conventional surface $CO_2$ observations, but they have
up to an order of magnitude higher single-sounding uncertainty and potential systematic errors
compared to the *in situ* and flask $CO_2$ observations. Recent progress in instrument error
characterization, spectroscopy, and retrieval methods have significantly improved the accuracy
and precision of the $X_{CO2}$ retrievals (O'Dell et al., 2018; Kiel et al., 2019). The single sounding
random error of $X_{CO2}$ from OCO-2 is ~1.0 ppm (Kulawik et al., 2019). A recent study by Byrne et
al. (2020) shows less than a 0.5 ppm difference between posterior $X_{CO2}$ constrained by a recent
data set, ACOS-GOSAT b7 $X_{CO2}$ retrievals, and those constrained by conventional surface $CO_2$
observations. Chevallier et al. (2019) also showed that an OCO-2 based flux inversion had similar
performance to surface $CO_2$ based flux inversions when comparing posterior $CO_2$ mole fractions
to aircraft $CO_2$ in the free troposphere. Results from these studies show that systematic
uncertainties in $CO_2$ retrievals from satellites are comparable to, or smaller than, other uncertainty
sources in atmospheric inversions (e.g. transport).

A newly-developed biogeochemical model-data fusion system, CARDAMOM, made progress in
producing NBE uncertainties, along with mean values that are consistent with a variety of
observations assimilated through a Markov Chain Monte Carlo (MCMC) method (Bloom et al.,
2016; 2020). Transport model errors in general have also been reduced relative to earlier transport
model intercomparison efforts, such as TransCom 3 (Gurney et al., 2004; Gaubert et al., 2019).
Advancements in satellite retrieval, transport, and prior terrestrial biosphere modeling have led to
more mature inversions constrained by satellite $X_{CO2}$ observations.

Two satellites, GOSAT and OCO-2, have now produced more than 10 years of observations. Here
we harness the CMS-Flux inversion framework (Liu et al., 2014; 2017; 2018; Bowman et al., 2017)
to generate an NBE product: CMS-Flux NBE 2020, by assimilating both GOSAT and OCO-2 from
2010–2018. The dataset is the longest satellite-constrained NBE product so far. The CMS-Flux
framework exploits globally available $X_{CO2}$ to infer spatially-resolved total surface-atmosphere
exchange. In combination with constituent fluxes, e.g., Gross Primary Production (GPP), NBE
from CMS-Flux framework have been used to assess the impacts of El Niño on terrestrial
biosphere fluxes (Bowman et al, 2017; Liu et al, 2017) and the role of droughts in the North
American carbon balance (Liu et al, 2018). These fluxes have furthermore been ingested into land-
surface data assimilation systems to quantify heterotrophic respiration (Konings et al., 2019),
evaluate structural and parametric uncertainty in carbon-climate models (Quetin et al., 2020), and
inform climate dynamics (Bloom et al., 2020). We present the regional NBE and its uncertainty
based on three types of regional masks: (1) latitude and continent, 2) distribution of biome types
(defined by plant functional types) and continent, and 3) TransCom regions (Gurney et al., 2004).

The outline of the paper is as follows: Section 2 describes methods, and Sections 3 and 4 describe
the dataset and the major NBE characteristics, respectively. We extensively evaluate the posterior
fluxes and uncertainties by comparing the posterior $CO_2$ mole fractions against aircraft
observations and the NOAA MBL reference $CO_2$, and a gross primary production (GPP) product
(section 5). In Section 6, we discuss the strength and weakness, and potential usage of the data. A
summary is provided in Section 7, and Section 8 describes the dataset availability and future plan.

**2   Methods**
**2.1 CMS-Flux inversion system**
The CMS-Flux framework is summarized in Figure 1. The center of the system is the CMS-Flux
inversion system, which optimizes NBE and air-sea net carbon exchanges with a 4D-Var inversion
system (Liu et al., 2014). In the current system, we assume no uncertainty in fossil fuel emissions,
which is a widely adopted assumption in global flux inversion systems (e.g., Crowell et al., 2019),
since the uncertainty in fossil fuel emissions at regional scales is substantially less than the NBE
uncertainties. The 4D-Var minimizes a cost function that includes two terms:
$$J(\mathbf{x}) = (\mathbf{x} - \mathbf{x}_b)^T \mathbf{B}^{-1}(\mathbf{x} - \mathbf{x}^b) + (\mathbf{y} - h(\mathbf{x}))^T \mathbf{R}^{-1}(\mathbf{y} - h(\mathbf{x})) \quad (1)$$
The first term measures the differences between the optimized fluxes and the prior fluxes
normalized by the prior flux error covariance **B**. The second term measures the differences between
observations ( $\mathbf{y}$ ) and the corresponding model simulations ( $h(\mathbf{x})$ ) normalized by the observation
error covariance **R**. The term $h(\cdot)$ is the observation operator that calculates observation-
equivalent model-simulated $X_{CO2}$. The 4D-Var uses the adjoint (i.e., the backward integration of
the transport model) (Henze et al., 2004) of the GEOS-Chem transport model to calculate the
sensitivity of the observations to surface fluxes. The configurations of the inversion system are
summarized in Table 1. We run both the forward and adjoint at 4° x 5° spatial resolution, and
optimize monthly NBE and air-sea carbon fluxes at each grid point from January 2010 to
December 2018. Inputs for the system include prior carbon fluxes, meteorological drivers, and the
satellite $X_{CO2}$ (Figure 1). Section 2.2 (Table 2) describes the prior flux and its uncertainties, and
section 2.3 (Table 3) describes the observations and the corresponding uncertainties.

**2.2 The prior $CO_2$ fluxes and uncertainties**
The prior $CO_2$ fluxes include NBE, air-sea carbon exchange, and fossil fuel emissions (see Table
2). The data sources for the prior fluxes are listed in Table 7 and provided in the gridded fluxes.
Methods to generate prior ocean carbon fluxes and fossil fuel emissions are documented in Brix
et al., (2015), Caroll et al. (2020), and Oda et al. (2018). The focus of this dataset is optimized
terrestrial biosphere fluxes, so we briefly describe the prior terrestrial biosphere fluxes and their
uncertainties.

We construct the NBE prior using the CARDAMOM framework (Bloom et al., 2016). The
CARDAMOM data assimilation system explicitly represents the time-resolved uncertainties in the
NBE. The prior estimates are already constrained with multiple data streams accounting for
measurement uncertainties following a Bayesian approach similar to that used in the 4D-
variational approach. We use the CARDAMOM setup as described by Bloom et al. (2016, 2020)
resolved at monthly timescales; data constraints include GOME-2 solar-induced fluorescence
(Joiner et al., 2013), MODIS Leaf Area Index (LAI), and biomass and soil carbon (details on the
data assimilation are provided in Bloom et al. (2020)). In addition, mean GPP and fire carbon
emissions from 2010 - 2017 are constrained by FLUXCOM RS+METEO version 1 GPP
(Tramontana et al., 2016; Jung et al., 2017) and GFEDv4.1s (Randerson et al., 2018), respectively,
both assimilated with an uncertainty of 20%. We use the Olsen and Randerson (2001) approach to
downscale monthly GPP and respiration fluxes to 3-hourly timescales, based on ERA-interim re-
analysis of global radiation and surface temperature. Fire fluxes are downscaled using the
GFEDv4.1 daily and diurnal scale factors on monthly emissions (Giglio et al., 2013).
Posterior CARDAMOM NBE estimates are then summarized as NBE mean and standard
deviation values.

The NBE from CARDAMOM shows net carbon uptake of 2.3 GtC/year over the tropics and close
to neutral in the extratropics (Figure B1). The year-to-year variability (i.e., interannual variability,
IAV) estimated from CARDAMOM from 2010 –2017 is generally less than 0.1 $gC/m^2$/day outside
of the tropics (Figure B1). Because of the weak interannual variability estimated by CARDAMOM,
we use the same 2017 NBE prior for 2018.

CARDAMOM generates uncertainty along with the mean state. The relative uncertainty over the
tropics is generally larger than 100%, and the magnitude is between 50% and 100% over the extra-
tropics (Figure B2). We assume no correlation in the prior flux errors in either space or time. The
temporal and spatial error correlation estimates can in principle be computed by CARDAMOM.
We anticipate incorporating these error correlations in subsequent versions of this dataset.

**2.3 Column $CO_2$ observations from GOSAT and OCO-2**
We use the satellite-column $CO_2$ retrievals from Atmospheric Carbon Observations from Space
(ACOS) team for both GOSAT (version 7.3) and OCO-2 (version 9) (Table 3). The use of the
same retrieval algorithm and validation strategy adopted by the ACOS team to process both
GOSAT and OCO-2 spectra maximizes the consistency between these two datasets. Both GOSAT
and OCO-2 satellites carry high-resolution spectrometers optimized to return high precision
measurements of reflected sunlight within $CO_2$ and $O_2$ absorption bands in the shortwave infrared
(Crisp et al., 2012). Both satellites fly in a sun-synchronous orbit. GOSAT has a 13:00 ± 0.15
hours local passing time and a three-day ground track repeat cycle. The footprint of GOSAT is
~10.5 km in diameter in sun-nadir view (Crisp et al., 2012). The daily number of soundings
processed by the ACOS-GOSAT retrieval algorithm is between a few hundreds to ~2000. Further
quality control and filtering reduce the ACOS-GOSAT $X_{CO2}$ retrievals to ~100 – 300 daily (Figure
B5 in Liu et al., 2017). We only assimilate ACOS-GOSAT land nadir observations flagged as
being good quality, which are the retrievals with quality flag equal to zero.

OCO-2 has a 13:30 local passing time and 16-day ground track repeat cycle. The nominal
footprints of the OCO-2 are 1.25 km wide and ~2.4 km along the orbit. Because of their small
footprints and sampling strategy, OCO-2 has many more $X_{CO2}$ retrievals than ACOS-GOSAT. To
reduce the sampling error due to the resolution differences between the transport model and OCO-
2 observations, we generate super observations by aggregating the observations within ~100 km
(along the same orbit) (Liu et al., 2017). The super-obing strategy was first proposed in numerical
weather prediction (NWP) to assimilate dense observations (Lorenc, 1981), and is still broadly
used in NWP (e.g., Liu and Rabier, 2003). More detailed information about OCO-2 super
observations can be found in Liu et al. (2017). OCO-2 has four observing modes: land nadir, land
glint, ocean glint, and target. Following Liu et al. (2017), we only use land nadir observations. The
super observations have more uniform spatial coverage and are more comparable to the spatial
representation of ACOS-GOSAT observations and the transport model (see Figure B5 in Liu et
al., 2017).

We directly use observational uncertainty provided with ACOS-GOSAT b7.3 to represent the
observation error statistics, **R**, in Eq 1. The uncertainty of the OCO-2 super observations is the
sum of the variability of $X_{CO2}$ used to generate each individual super observation and the mean
uncertainty provided in the original OCO-2 retrievals. Kulawik et al. (2019) showed that both
OCO-2 and ACOS-GOSAT bias-corrected retrievals have a mean bias of -0.1 ppm when compared
with $X_{CO2}$ from Total Carbon Column Observing Network (TCCON) (Wunch et al., 2011),
indicating consistency between ACOS-GOSAT and OCO-2 retrievals. O'Dell et al. (2018) showed
that the OCO-2 $X_{CO2}$ land nadir retrievals has RMS error of ~1.1 ppm when compared to TCCON
retrievals; the differences between OCO-2 $X_{CO2}$ retrievals and surface $CO_2$ constrained model
simulations are well within 1.0 ppm over most of the locations in the Northern Hemisphere (NH),
where most of the surface $CO_2$ observations are located.

The magnitude of observation errors used in **R** is generally above 1.0 ppm, larger than the sum of
random error and biases in the observations. The ACOS-GOSAT b7.3 observations from July
2009–June 2015 are used to optimize fluxes between 2010 and 2014, and the OCO-2 $X_{CO2}$
observations from Sep 2014–June 2019 are used to optimize fluxes between 2015 and 2018.

The observational coverage of ACOS-GOSAT and OCO-2 is spatiotemporally dependent, with
more coverage during summer than winter over the NH, and more observations over mid-latitudes
than over the tropics (Figure B3). The variability (i.e., standard deviation) of annual total number
of observations from 2010–2014 is within 4% of the annual mean number for ACOS-GOSAT.
Except for a data gap in 2017 caused by a malfunction of the OCO-2 instrument, the variability of
the annual total number of observations between 2015 and 2018 is within 8% of the annual mean
number for OCO-2.

**2.4 Uncertainty quantification**
The posterior flux error covariance is the inverse Hessian, which incorporates the transport,
measurement, and background errors at the 4D-Var solution (Eq. 13 in Bowman et al, 2017).
Posterior flux uncertainty projected to regions can be estimated analytically based on the methods
described in Fisher and Courtier (1995) and Meirink et al. (2008), using either flux singular vectors
or flux increments obtained during the iterative optimization (e.g., Niwa and Fujii, 2020). In this
study, we rely on a Monte Carlo approach to quantify posterior flux uncertainties following
Chevallier et al. (2010) and Liu et al. (2014), which is simpler and widely used. In this approach,
an ensemble of flux inversions is carried out with an ensemble of priors and simulated observations
to sample the uncertainties of prior fluxes (i.e., $\mathbf{B}$ in eq. 1) and observations ($\mathbf{R}$ in Eq. 1),
respectively. The magnitude of posterior flux uncertainties is a function of assumed uncertainties
in prior fluxes and observations, as well as the density of observations. Since the density of
GOSAT and OCO-2 observations are stable (section 2.3) within their respective data record, we
characterize the posterior flux uncertainties for 2010 and 2015 only, and assume the flux
uncertainties for 2011–2014 are the same as 2010 and flux uncertainties for 2016–2018 are the
same as 2015.

**2.5 Evaluation of posterior fluxes**
Direct NBE estimates from flux towers only provide a spatial representation of roughly 1 – 3
kilometers (Running et al., 1999), not appropriate to evaluate regional NBE from top-down flux
inversions. Thus, we use two methods to indirectly evaluate the posterior NBE and its uncertainties.
One is to compare annual NBE anomalies and seasonal cycle to a gross primary production (GPP)
product. The other is to compare posterior $CO_2$ mole fractions to independent (i.e., not assimilated
in the inversion) aircraft and the NOAA MBL reference observations. The second method has been
broadly used to indirectly evaluate posterior fluxes from top-down flux inversions (e.g., Stephens
et al., 2007; Liu and Bowman, 2016; Chevallier et al., 2019; Crowell et al., 2019). In addition to
these two methods, we also compare the NBE seasonal cycles to three publicly available top-down
NBE estimates that are constrained by surface $CO_2$ observations (Tables 3 and 7).
**2.5.1 Evaluation against independent gross primary production (GPP) product**
NBE is a small residual difference between two large terms: total ecosystem respiration (TER)
and GPP, plus fire. A positive NBE anomaly (i.e., less uptake from the atmosphere) has been
shown to correspond to reduced GPP caused by climate anomalies (e.g., Bastos et al., 2018), and
the magnitude of net uptake is proportional to GPP in most biomes observed by flux tower
observations (e.g., Falk et al., 2008). Since NBE is related not only to GPP, the comparison to GPP
only serves as a qualitative measure of the NBE quality. For example, we would expect that the
posterior NBE seasonality to be anti-correlated with GPP in the temperate and high latitudes. In
this study, we use FLUXSAT GPP (Joiner et al., 2018), which is an upscaled GPP product based
on flux tower GPP observations and satellite-based geometry adjusted reflectance from the
MODerate-resolution Imaging Spectroradiometer (MODIS) and solar-induced chlorophyll
fluorescence observations from Global Ozone Monitoring Experiment – 2 (GOME-2) (Joiner et
al., 2013). Joiner et al. (2018) show that the agreement between FLUXSAT-GPP and GPP from
flux towers is better than other available upscaled GPP products.
**2.5.2 Evaluation against aircraft and the NOAA marine boundary layer (MBL)**
**reference $CO_2$ observations**
The aircraft observations used in this study include those published in OCO-2 MIP ObsPack
August 2019 (CarbonTracker team, 2019), which include regular vertical profiles from flask
samples collected on light aircraft by NOAA (Sweeney et al., 2015) and other laboratories, regular
(two to four weekly) vertical profiles from the Instituto de Pesquisas Espaciais (INPE) over
tropical South America (SA) (Gatti et al., 2014), and from the Atmospheric Tomography (ATom,
Wofsy et al., 2018), HIAPER Pole-to-Pole (HIPPO, Wofsy et al., 2011), the $O_2/N_2$ Ratio and $CO_2$
airborne Southern Ocean Study (ORCAS)  (Stephens et al., 2017), and Atmospheric Carbon and
Transport - America (ACT-America, Davis et al., 2018) aircraft campaigns (Table 3). Figure 2
shows the aircraft observation coverage and density between 2010 and 2018. Most of the aircraft
observations are concentrated over NA. ATom had four (1–4) campaigns between August 2016 to
May 2018, spanning four seasons over the Pacific and Atlantic Ocean. HIPPO had five (1–5)
campaigns over the Pacific, but only HIPPO 3–5 occurred between 2010 and 2011. HIPPO 1–2
occurred in 2009. Based on the spatial distribution of aircraft observations, we divide the
comparison into nine regions: Alaska, mid-latitude NA, Europe, East Asia, South Asia, Africa,
Australia, Southern Ocean, and South America (Table 4 and Figure 2).

We calculate several quantities to evaluate the posterior fluxes and their uncertainty with aircraft
observations. One is the monthly mean differences between posterior and aircraft $CO_2$ mole
fractions. The second is the monthly root mean square errors (RMSE) over each of nine sub-
regions, which is defined as:
$RMSE = (\frac{1}{n}\sum_{i=1}^{n}(y^o_{aircraft} - y^b_{aircraft})^2_i)^{\frac{1}{2}}$ (2)
where $y^o_{aircraft}$ is the $i$th aircraft observation,  $y^b_{aircraft}$ is the corresponding posterior $CO_2$ mole
fraction sampled at the $i$th aircraft location, and $n$ is the number of aircraft observations over each
region. The RMSE is computed over the *n* aircraft observations within one of the nine sub-regions.
The mean differences indicate the magnitude of the mean posterior $CO_2$ bias, while the RMSE
includes both random and systematic errors in posterior $CO_2$. The bias and RMSE could be due to
errors in posterior fluxes, transport, and initial $CO_2$ concentrations. When errors in transport and
initial $CO_2$ concentrations are smaller than the errors in the posterior fluxes, the magnitude of
biases and *RMSE* indicates the accuracy of the posterior fluxes.

To evaluate the magnitude of posterior flux uncertainty estimates, we compare *RMSE* against the
standard deviation of ensemble simulated aircraft observations (equation 3) from the Monte Carlo
method (*RMSE_{MC}*). The quantity $RMSE_{MC}$ can be written as:
$RMSE_{MC} = [\frac{1}{nens}\sum_{iens=1}^{nens}((y_{aircraft}^{b(MC)})_{iens} - \bar{y}_{aircraft}^{b(MC)})^2 ]^{\frac{1}{2}}$ (3)
The variable $(y_{aircraft}^{b(MC)})_{iens}$ is the $i^{th}$ ensemble member of simulated aircraft observations from
Monte Carlo ensemble simulations, $\bar{y}_{aircraft}^{b(MC)}$ is the mean, and *nens* is the total number of ensemble
members. For simplicity, in equation (3), we drop the indices for the aircraft observations used in
equation (2). In the absence of errors in transport and initial $CO_2$ concentrations, when the
estimated posterior flux uncertainty reflects the "*true*" posterior flux uncertainty, we show in the
*Appendix* that:
$RMSE^2 = \frac{1}{n}\sum_{i=1}^{n} R_{i,i} + RMSE_{MC}^2$    (4)
where $R_{aircraft}$ is the aircraft observation error variance, which could be neglected on regional
scale.

We further calculate the ratio *r* between *RMSE* and $RMSE_{MC}$:
$r = \dfrac{RMSE}{RMSE_{MC}}$            (5)
A ratio close to one indicates that the posterior flux uncertainty reflects the true uncertainty in the
posterior fluxes when the transport errors are small.

The presence of transport errors will make the comparison between $RMSE$ and $RMSE_{MC}$
potentially difficult to interpret. Even when $RMSE_{MC}$ represents the actual uncertainty in posterior
fluxes, the $RMSE$ could be larger than $RMSE_{MC}$ , since the differences between aircraft
observations and model simulated posterior mole fractions $RMSE$ could be due to errors in both
transport and the posterior fluxes, while $RMSE_{MC}$ only reflects the impact of posterior flux
uncertainty on simulated aircraft observations. In this study, we assume the primary sources of
$RMSE$ come from errors in posterior fluxes.

The $RMSE$ and $RMSE_{MC}$ comparison only shows differences in $CO_2$ space. We further calculate
the sensitivity of the $RMSE$ to the posterior flux using the GEOS-Chem adjoint. We first define a
cost function $J$ as:
$J = RMSE^2$      (6)
The sensitivity of the mean-square error to a flux, $x$, at location $i$ and month $j$ is
$w_{i,j} = \dfrac{\partial J}{\partial x_{i,j}} \times x_{i,j}$    (7)
This sensitivity is normalized by the flux magnitude. Equation 7 can be interpreted as the
sensitivity of the $RMSE^2$ to a fractional change in the fluxes. We can estimate the time-integrated
magnitude of the sensitivity over the entire assimilation window by calculating:
$S_i = \dfrac{\sum_{j=1}^{M} |w_{i,j}|}{\sum_{k=1}^{P} \sum_{j=1}^{M} |w_{k,j}|}$    (8)
where $P$ is the total number of grid points and $M$ is the total number of months from the time of
the aircraft data to the beginning of the inversion. The numerator of equation (8) quantifies the
absolute total sensitivity of the $RMSE^2$ to the fluxes at the $i^{th}$ grid. Normalized by the total absolute
sensitivity across the globe, the quantity $S_i$ indicates the relative sensitivity of $RMSE^2$ to fluxes at
the $i^{th}$ grid point. Note that $S_i$ is unitless, and it only quantifies sensitivity, not the contribution of
fluxes at each grid to $RMSE^2$.

We use the NOAA MBL reference dataset (Table 7) to evaluate the $CO_2$ seasonal cycle over four
latitude bands: 90ºN-60ºN, 60ºN-20ºN, 20ºN-20ºS, and 20ºS-90ºS. The MBL reference is based
on a subset of sites from the NOAA Cooperative Global Air Sampling Network. Only
measurements that are representative of a large volume air over a broad region are considered. In
the comparison, we first remove the global mean $CO_2$
(https://www.esrl.noaa.gov/gmd/ccgg/trends/global.html ) from both the NOAA MBL reference
and the posterior $CO_2$.

**2.6 Regional masks**
We provide posterior NBE from 2010 – 2018 using three sets of regional masks (Figure 3), in
addition to the gridded product. The regional mask in Figure 3A is based on a combination of
seven plant function types condensed from MODIS IGBP and the TransCom -3 regions (Gurney
et al., 2004), which is referred as Region Mask 1 (RM1) in later description. There are 28 regions
in Figure 3A: six in NA, four in SA, five in Eurasia (north of 40˚N), three in tropical Asia, three
in Australia, and seven in Africa. The regional mask in Figure 3B is based on latitude and
continents with 13 regions in total, which is referred as Region Mask 2 (RM2) in later description.
Figure 3C is the TransCom regional mask with 11 regions on land.

**3 Dataset description**
We present the fluxes as globally, latitudinally, and regionally aggregated time series. We show
the nine-year average fluxes aggregated into RM1, RM2, and TransCom regions (Figure 3). The
aggregations are geographic (latitude and continent) and bio-climatic (biome by continent).  For
each region in the geographic and biome aggregations, we show nine-year mean annual net fluxes
and uncertainties, and then the annual fluxes for each region as a set of time-series plots. The
month-by-month fluxes and uncertainties are available in tabular format, so the actual aggregated
fluxes may be readily compared to bottom-up extrapolated fluxes and Earth System models. Users
can also aggregate the gridded fluxes and uncertainties based on their own defined regional masks.
Table 5 provides a complete list of all data products available in the dataset. In section 4, we
describe the major characteristics of the dataset.
**4 Characteristics of the dataset**
**4.1 Global fluxes**
The annual atmospheric $CO_2$ growth rate, which is the net difference between fossil fuel emissions
and total annual sink over land and ocean, is well-observed by the NOAA surface $CO_2$ observing
network (https://www.esrl.noaa.gov/gmd/ccgg/ggrn.php). We compare the global total flux estimates
constrained by GOSAT and OCO-2 with the NOAA $CO_2$ growth rate from 2010–2018, and discuss
the mean carbon sink over land and ocean. Over these nine years, the satellite-constrained
atmospheric $CO_2$ growth rate agrees with the NOAA observed $CO_2$ growth rate within the
uncertainty of the posterior fluxes (Figure 4). The mean annual global surface $CO_2$ fluxes (in Gt
C/yr) are derived from the NOAA observed $CO_2$ growth rate (in ppm/yr) using a conversion factor
of 2.124 GtC/ppm (Le Quéré et al., 2018). The estimated growth rate has the largest discrepancy
with the NOAA observed growth rate in 2014, which may be due to a failure of one of the two
solar paddles of GOSAT in May 2014 (Kuze et al., 2016). Over the nine years, the estimated total
accumulated carbon in the atmosphere is 41.5 ± 2.4 GtC, which is slightly lower than the
accumulated carbon based on the NOAA $CO_2$ growth rate (45.2 ± 0.4 GtC). On average, we
estimate that the NBE is 2.0 ± 0.7 GtC, ~20 ± 8% of fossil fuel emissions, and the ocean sink is
3.0 ± 0.1 GtC, ~ 30 ± 1% of fossil fuel emissions (Figure 4). These numbers are within the ranges
of the corresponding GCB estimates from Freidlingstein et al., 2019 (referred as GCB-2019
hereafter). The mean NBE and ocean sink from GCB-2019 are 2.0 ± 1.0 GtC and 2.5 ± 0.5 GtC
respectively, which are 21 ± 10%  and 26 ± 5% of fossil fuel emissions respectively between 2010–
2018. The GCB does not report NBE directly, we calculate NBE from GCB-2019 as the residual
differences between fossil fuel, ocean net carbon sink, and atmospheric $CO_2$ growth rate. It is also
equivalent to ($S_{LAND}$ + $B_{IM}$ - $E_{LUC}$) reported by Freidlingstein et al., 2019, where $S_{LAND}$ is terrestrial
sink, $B_{IM}$ is a budget imbalance, and $E_{LUC}$ is land use change.   Over these nine years, we estimate
that NBE ranges from 3.6 GtC (~37% of fossil fuel emissions) in 2011 (a La Niña year), to only
0.5 GtC, (~5% of fossil fuel emissions) in 2015 (an El Niño year), consistent with 3.3 GtC (35%
of fossil fuel) in 2011 to 0.9 GtC (7% of fossil fuel) in 2015 estimated from GCB-2019. We
estimate that the ocean sinks range from 3.5 GtC in 2015 to 2.3 GtC in 2012, larger than the
estimated ocean flux ranges of 2.7 in 2016 to 2.5 in 2012 reported by Freidlingstein et al. (2019).
**4.2 Mean regional fluxes and uncertainties**
Figure 5 shows the nine-year mean regional annual fluxes, uncertainty, and its variability between
2010–2018. Table 6 shows an example of the dataset corresponding to Figure 5 A, D, and G. It

shows that large net carbon uptake occurs over Eurasia, NA, and the Southern Hemisphere (SH)

mid-latitudes. The largest net carbon uptake is over the eastern US (-0.4 ± 0.1 GtC (1σ uncertainty))

and high latitude Eurasia (-0.5 ± 0.1 GtC) (Figure 5A, B). We estimate a net land carbon sink of

2.5 ± 0.3 GtC/year between 2010–2013 over the NH mid to high latitudes, which agrees with 2.4

± 0.6 GtC estimates over the same time periods based on a two-box model (Ciais et al., 2019). Net

uptake in the tropics ranges from close-to-neutral in tropical South America (0.1 ± 0.1 GtC) to a

net source in northern Africa (0.6 ± 0.2 GtC) (Figure 5A, B). The tropics exhibit both large

uncertainty and large variability. The NBE interannual variability over northern Africa and tropical

SA are 0.5 GtC and 0.3 GtC respectively, larger than the 0.2 GtC and 0.1 GtC uncertainty (Figure

5D, E). We also find collocation of regions with large NBE and FLUXSAT-GPP interannual

variability (Figure B4). The availability of flux estimates over the broadly used TransCom regions

make it easy to compare to previous studies. For example, we estimate that the annual net carbon

uptake over North America is 0.7 ± 0.1 GtC/year with 0.2 GtC variability between 2010 and 2018,

which agrees with 0.7 ± 0.5 GtC/year estimates based on surface $CO_2$ observations between 1996-

2007 (Peylin et al., 2013).

## 4.3 Interannual variabilities and uncertainties

Here we present hemispheric and regional NBE interannual variabilities and corresponding

uncertainties (Figures 6 and 7, and corresponding tabular data files). In Figure 6, we further divide

the globe into three large latitude bands: tropics (20°S–20°N), NH extra-tropics (20°N–85°N), and

SH extra-tropics (60°S–20°S). The tropical NBE contributes 90% to the global NBE interannual

variability (IAV). The IAV of NBE over the extra-tropics is only about one-third of that over the

tropics. The dominant role of tropical NBE in the global IAV of NBE agrees with Figure 4 in

Sellers et al. (2018). The top-down global annual NBE anomaly is within the 1.0 GtC/yr
uncertainty of residual NBE (i.e., fossil fuel – atmospheric growth – ocean sink) calculated from
GCB-2019 (Friedlinston et al., 2019) (Figure 6).

Figure 7 shows the annual NBE anomalies and uncertainties over a few selected regions based on
RM1. Positive NBE indicates reduced net uptake relative to the 2010–2018 mean, and vice versa.
Also shown in Figure 7 are GPP anomalies estimated from FLUXSAT. Positive GPP indicates
increased productivity, and vice versa. GPP drives NBE in years where anomalies are inversely
correlated (e.g., positive NBE and negative GPP), and TER drives NBE in years where anomalies
of GPP and NBE have the same sign or are weakly correlated. Over tropical SA evergreen
broadleaf forest, the largest positive NBE anomalies occur during the 2015–2016 El Niño,
corresponding to large reductions in productively, consistent with Liu et al. (2017). In 2017, the
region sees increased net uptake and increased productivity, implying a recovery from the 2015–
2016 El Niño event. The variability in GPP explains 80% of NBE variability over this region over
the nine-year period. In Australian shrubland, our inversion captures the increased net uptake in
2010 and 2011 due to increased precipitation (Poulter et al., 2014) and increased productivity. The
variability in GPP explains 70% of the interannual variability in NBE. Over tropical south America
savanna, the NBE interannual variability also shows strong negative correlations with GPP, with
GPP explaining 40% of NBE interannual variability. Over the mid-latitude regions where the IAV
is small, the $R^2$ between GPP and NBE is also small (0.0–0.5) as expected. But the increased net
uptake generally corresponds to increased productivity. We also do not expect perfect negative
correlation between NBE anomalies and GPP anomalies, as discussed in section 2.5. The
comparison between NBE and GPP provides insight into when and where net fluxes are likely
dominated by productivity.

**4.4 Seasonal cycle**
We provide the regional mean NBE seasonal cycle, its variability, and uncertainty based on the
three regional masks (Table 5). Here we briefly describe the characteristics of the NBE seasonal
cycle over the 11 TransCom regions, and its comparison to three independent top-down inversion
results based on surface $CO_2$, which are CT-Europe (e.g., van der Laan-Luijkx et al., 2017) CAMS
(Chevallier et al., 2005), and Jena CarbonScope (Rödenbeck et al., 2003). CMS-Flux-NBE differs the
most from surface-$CO_2$ based inversions over the South American Tropical, Northern Africa,
tropical Asia, and NH boreal regions. The CMS-Flux NBE has a larger seasonal cycle amplitude
over tropical Asia and Northern Africa, where the surface $CO_2$ constraint is weak, while it has a
smaller seasonal cycle amplitude over the boreal region; this may be due to the sparse satellite
observations over the high latitudes and weaker seasonal amplitude of the prior CARDAMOM
fluxes. The comparison to FluxSat GPP can only qualitatively evaluate the NBE seasonal cycle,
but cannot differentiate among different estimates. In general, the months that have larger
productivity corresponds to months with a net uptake of carbon from the atmosphere, especially
over the NH (Figure 8). More research is still needed to understand the  seasonal cycles of NBE,
including its phase (i.e., transition from source to sink) and amplitude (peak-to-trough difference),
and its relationships with GPP and respiration.

**5    Evaluation against independent aircraft $CO_2$ observations**
**5.1 Comparison to aircraft observations over nine sub-regions**
In this section, we evaluate posterior $CO_2$ against aircraft observations over the nine sub-regions
listed in Table 4 and Figure 2. We compare the posterior $CO_2$ to aircraft $CO_2$ mole fractions above
the planetary boundary layer and up to mid troposphere (1–5 km) at the locations and time of
aircraft observations, and then calculate the monthly mean error statistics between 1–5 km. The
aircraft observations between 1–5 km are more sensitive to regional fluxes (Liu et al., 2015; Liu
and Bowman, 2016). Scatter plots in the left column of Figure 9 show regional monthly mean de-
trended aircraft $CO_2$ observations (x-axis) versus the simulated detrended posterior $CO_2$ (y-axis).
We used the NOAA global $CO_2$ trend to detrend both the observations and model simulated mole
fractions (ftp://aftp.cmdl.noaa.gov/products/trends/co2/co2_trend_gl.txt). Over the NH regions (A,
B, C, D) and Africa (F), the $R^2$ is greater than or equal to 0.9, which indicates that the posterior
$CO_2$ captures the observed seasonality. The low $R^2$ (0.7) value in South Asia is caused by one
outlier. Over the Southern Ocean, Australia, and SA, the $R^2$ is between 0.2 and 0.4, reflecting
weaker $CO_2$ seasonality over these regions and possible bias in ocean flux estimates (see
discussions later).

The right panel of Figure 9 shows the monthly mean differences between posterior $CO_2$ and aircraft
observations (black), *RMSE* (equation 2) (blue line), and *RMSE$_{MC}$* (equation 3) (red line). The
magnitude of the mean differences between the posterior $CO_2$ and aircraft observations is less than
0.5 ppm except over the Southern Ocean, which has a -0.8 ppm bias. The mean differences between
posterior $CO_2$ and aircraft observations are primarily caused by errors in transport and biases in
assimilated satellite observations, while *RMSE$_{MC}$* is 'internal flux error' projected into mole
fraction space. With the exception of the Southern Ocean, for all regions mean bias is significantly
less than *RMSE$_{MC}$*, which suggests that transport and data bias in satellite observations may be
much smaller than the internal flux errors. Note that $RMSE_{MC}$ is smaller than $RMSE$ over the first
~six months of simulation, which may indicate a dominant impact of errors in transport and initial
$CO_2$ concentration on posterior $CO_2$ $RMSE$.

As demonstrated in section 2.5, comparing $RMSE$ and $RMSE_{MC}$ is a test of the accuracy of posterior
flux uncertainty estimate. Over all the regions, the differences between $RMSE$ and $RMSE_{MC}$ are
smaller than 0.3 ppm, which indicates a comparable magnitude between empirical posterior flux
uncertainty estimates from the Monte Carlo method and the actual posterior flux uncertainty over
the regions that these aircraft observations are sensitive to. These aircraft observations are sensitive
to NBE over a broad region as shown in Figure B5. Note, Figure B5 and Figures B8-B10 are
calculated using equation (8).

**5.2 Comparison to aircraft observations from ATom and HIPPO aircraft campaigns**
Figures 10 and 11 show comparisons to aircraft $CO_2$ from ATom 1–4 campaigns spanning four
seasons, and HIPPO 3–5 over the Pacific Ocean between 1–5 km. The vertical curtain comparisons
are shown in Figure B6 and B7. The mean differences between posterior $CO_2$ and aircraft $CO_2$ are
quite uniform (within 0.5 ppm) throughout the column except over the Atlantic Ocean during
ATom 1–2 and the Southern Ocean during ATom 1 (Figures S6 and S7). Also shown in Figures
10 and 11 are $RMSE$ of each aircraft campaign (middle column) and the ratio between $RMSE$ and
$RMSE_{MC}$ (right column). A ratio larger than one between $RMSE$ and $RMSE_{MC}$ indicates errors in
either transport or underestimation of the posterior flux uncertainty (section 2.5).

Over most of the flight tracks during ATom 1–4, the posterior $CO_2$ errors are between -0.5 and 0.5
ppm, the *RMSE* is smaller than 0.5 ppm, and the ratio between *RMSE* and $RMSE_{MC}$ is smaller than
or equal to 1. However, off the coast of Africa during ATOM -1 and -2 and over the Southern
Ocean during ATOM-1, the mean differences between posterior $CO_2$ and aircraft observations are
larger than 0.5 ppm. During ATOM-1 (29 July – 23 Aug 2016), the mean differences between
posterior $CO_2$ and aircraft $CO_2$ show large negative biases, while during ATOM-2 (26 Jan 2017–
21 Feb 2017), it has large positive biases off the coast of Africa. The ratio between *RMSE* and
$RMSE_{MC}$ is significantly larger than one over these regions, which indicates an underestimation of
posterior flux uncertainty or large magnitude of transport errors during that time period.

We further run adjoint sensitivity analyses over the three regions with ratios significantly larger
than one to identify the posterior fluxes that could contribute to the large differences between
posterior $CO_2$ and aircraft observations during ATOM 1–2. We run the adjoint model backward
for three months from the observation time and calculate $S_i$ as defined in equation (7). The adjoint
sensitivity analysis indicates that the large mismatch between aircraft observations and model
simulations during ATOM-1 and -2 off the coast of Africa could be potentially driven by errors in
posterior fluxes over tropical Africa (Figure B8). The large posterior $CO_2$ errors and large ratio
between *RMSE* and $RMSE_{MC}$ over the Southern Ocean during ATOM-1 are driven by flux errors
in oceanic fluxes around 30°S and over Australia (Figure B9), which also contribute to the large
errors in comparison to aircraft observations over the Southern Ocean shown in Figure 9 H.

During the HIPPO aircraft campaigns, the absolute errors in posterior $CO_2$ across the Pacific are
less than 0.5 ppm except over the Arctic Ocean and over Alaska in summer (Figure 11), consistent
with Figure 10A. The large errors over the Arctic Ocean may be related to both transport errors
and the accuracy of high latitude fluxes. Byrne et al. (2020) provide a brief summary of the
challenges in simulating $CO_2$ over high latitudes using a transport model with 4° x 5° resolution.
Increasing the resolution of the transport model may reduce transport errors over high latitudes.

We run adjoint sensitivity analysis over the high-latitude regions where the differences between
posterior $CO_2$ and aircraft observations are large (Figure 11). The adjoint sensitivity analysis
(Figure B10) shows that the large errors over these regions could be driven by errors in fluxes over
Alaska as well as broad NH mid-latitude regions.

**5.3 Comparison to MBL reference sites**
Since MBL reference sites sample air over broad regions, the comparison to detrended MBL
observations indirectly evaluates the NBE over large regions. Figure 12 shows the comparison
over four latitude bands. The uncertainty of posterior $CO_2$ concentration is from the MC method.
Except over 90°S-20°S, the differences between observations and posterior $CO_2$ are within
posterior $CO_2$ uncertainty estimates. The posterior $CO_2$ concentrations have the smallest bias and
random errors over the tropical latitude band. The $R^2$ is above 0.9 over NH mid to high latitudes,
consistent with Figure 9. Over 90°S-20°S, the posterior $CO_2$ has positive bias in 2013 and 2014
and negative bias and much weaker seasonality between Jan 2015 – Dec 2018 compared to
observations, which indicates possible biases in Southern Ocean flux estimates (Figure B11). The
low bias over the Southern Ocean is consistent with aircraft comparison during OCO-2 period
(Figures 9-10, Figure B9). The changes of performance after 2013 over 90°S-20°S is most likely
due to the prior ocean carbon fluxes. Evaluation of ocean carbon fluxes is out of scope of this study.
Note, since we only assimilate land-nadir $X_{CO2}$ observations in this study due to known issues with
the OCO-2 v9 ocean glint observations (O'Dell et all., 2018), the constraint of top-down inversion
on air-sea $CO_2$ exchanges is weak (not shown). The ocean glint observations of OCO-2 v10
observations have been improved compared to v9 (Osterman et al., 2020). We expect to have better
estimate of ocean carbon fluxes over the Southern Ocean when assimilating both land and ocean
$X_{CO2}$ observations from GOSAT and OCO-2 in the future.
**6   Discussion**
Evaluation of posterior flux uncertainty estimates by comparing posterior $CO_2$ error statistics
(*RMSE,* Equation 2) with the standard deviation of ensemble simulated $CO_2$ from Monte Carlo
uncertainty quantification method (*RMSE$_{MC}$*, equation 3) has its limitations. A comparable *RMSE*
and *RMSE$_{MC}$* indicates a small magnitude of transport errors and reasonable posterior uncertainty
estimates. A much larger *RMSE* than *RMSE$_{MC}$* could be due to errors in either transport or
underestimation of the posterior flux uncertainty or both. The presence of transport errors makes
the interpretation of the *RMSE* and *RMSE$_{MC}$* complex. A better, independent quantification of
transport errors is needed in the future in order to rigorously use the comparison statistics between
aircraft observations and posterior $CO_2$ to diagnose flux errors.

Comparison to aircraft observations shows regionally-dependent accuracy in posterior fluxes.
ATom observations show seasonally-dependent biases over the Atlantic, implying possible
seasonally dependent errors in posterior fluxes over northern to central Africa. Therefore, we
recommend combining NBE with other ancillary variables, e.g., GPP, to better understand carbon
dynamics. Combining NBE with component carbon fluxes can shed light on the processes
controlling the changes of NBE (e.g., Bowman et al, 2017; Liu et al., 2017). NBE can be written
as:
NBE= TER + fire - GPP   (8)
where TER is total ecosystem respiration (TER) (Figure 1). Satellite carbon monoxide (CO)
observations provide constraints on fire emissions (Arellano et al, 2006, van der Werf, 2008; Jones
et al, 2009; Jiang et al., 2015, Bowman et al, 2017; Liu et al., 2017). In addition to the FLUXSAT-
GPP product used here, solar induced chlorophyll fluorescence (SIF) can be directly used as a
proxy for GPP (e.g., Parazoo et al, 2014). Once NBE, fire, and GPP carbon fluxes are quantified,
TER can be calculated as a residual (e.g., Bowman et al, 2017; Liu et al., 2017, 2018).

Because of the diffusive manner of atmospheric transport and the limited observation coverage,
the gridded flux values are not independent from each other. The errors and uncertainties of the
fluxes at each individual grid point are larger than regional aggregated fluxes. Interpreting NBE at
each individual grid point requires caution. But at the same time, satellite $CO_2$ constrained NBE
can potentially resolve fluxes at spatial scales smaller than the traditional TransCom regions. Here,
we provide regional fluxes at two predefined regions in addition to TransCom. We encourage data
users to use the data at appropriate regional scales.

The variability and changes are more robust than the mean NBE fluxes from top-down flux
inversions in general (Baker et al., 2006b). The errors in transport and potential biases in
observations are mostly stable in time, so biases in the mean fluxes tend to cancel out when
computing interannual variability and year-to-year changes (Schuh et al., 2019; Crowell et al.,

618    2019).


The global fossil fuel emissions have ~5% uncertainty (GCB-2019). However, they are regionally
inhomogeneous. We neglect the uncertainties in fossil fuel emissions, which will introduce
additional error in regions of rapid fossil fuel growth or in areas with noisier statistics (Yin et al.,
2019). In the future, we will account for uncertainties in fossil fuel emissions.

The posterior NBE includes all types of land fluxes except fossil fuel emissions, which is
equivalent to the sum of land use change fluxes, land sinks, and residual imbalance published by
the GCB-2019. The sum of regional NBE and fossil fuel emissions is an index of the contribution
of any specific region to the changes of the atmospheric $CO_2$ growth rate. Since the predicted
changes of NBE in the future have large uncertainties (Lovenduski and Bonan, 2017), quantifying
regional NBE is critical to monitoring regional contributions to atmospheric $CO_2$ growth rate, and
ultimately to guide mitigation to limit warming to 1.5°C above pre-industrial levels (IPCC, 2018).

**7   Summary**
Terrestrial biosphere carbon fluxes are the largest contributor to the interannual variability of the
atmospheric $CO_2$ growth rate. Therefore, monitoring its change at regional scales is essential for
understanding how it responds to $CO_2$, climate and land use. Here, we present the longest terrestrial
flux estimates and their uncertainties constrained by $X_{CO2}$ from 2010–2018 on self-consistent
global and regional scales (CMS-Flux NBE 2020). We qualitatively evaluate the NBE estimates
by comparing its variability with GPP variability, and provide comprehensive evaluation of
posterior fluxes and the uncertainties by comparing posterior $CO_2$ with independent $CO_2$
observations from aircraft and the NOAA MBL reference sites. This dataset can be used in
understanding controls on regional NBE interannual variability, evaluating biogeochemical
models, and supporting the monitoring of regional contributions to changes in atmospheric $CO_2$.

**8    Data availability and future update**
The CMS-Flux NBE 2020 data are available at:  https://doi.org/10.25966/4v02-c391 (Liu et al.,
2020). The regional aggregated fluxes are provided as *csv* files with file size ~10MB, and the
gridded data is provided in NetCDF format with file size ~1.4 GB. The full ensemble of posterior
fluxes used to estimate posterior flux uncertainties are provided in NetCDF format with file size
~30MB. Table 7 lists the sources of the data used in producing and evaluating the CMS-Flux NBE
2020 data product.

The quality of $X_{CO2}$ from satellite observations is continually improving. The OCO-2 v10 $X_{CO2}$
has been released in June 2020 along with the full GOSAT record (June 2009–Jan 2020) processed
by the same retrieval algorithm as OCO-2. Continuing to improving the quality of satellite
observations and extending the NBE estimates beyond 2018 in the future will help us better
understand interactions between terrestrial biosphere carbon cycle and climate and provide support
in monitoring the regional contributions to the changes of atmospheric $CO_2$. Thus, we plan a future
update of the dataset on an annual basis, with a goal to support current scientific research and
policy making.
**9    Author contributions**
JL designed the study and led the writing of the paper in close collaboration with KB and DS. LB
helped generate the plots and created all the data files. AAB provided the prior of the terrestrial
biosphere carbon fluxes. NP helped interpret the GPP evaluation. DM and DC generated the prior
ocean carbon fluxes. TO generated the ODIAC fossil fuel emissions. JJ provided the FLUXSAT
GPP product. BD and SW provided and contributed to the interpretation of HIPPO aircraft $CO_2$
observation comparisons. BS, KM, and CS provided ORCAS aircraft $CO_2$ observations and
contributed interpretation of aircraft $CO_2$ observation comparisons. LVG and JM provided INPE
aircraft $CO_2$ observations and contributed interpretation of aircraft $CO_2$ observation comparisons.
CS and KM provided ATom and the NOAA aircraft CO2 observations and contributed
interpretation of aircraft $CO_2$ observation comparisons. We furthermore acknowledge funding
from the EU for the ERC project "ASICA" (grant number 649087) to Wouter Peters (Groningen
University) and EU and NERC (UK) funding to Emanuel Gloor (University of Leeds), which
contributed to the INPE Amazon greenhouse sampling program. All authors contributed to the
writing, and have reviewed and approved the paper.
**10   Competing interest**
The authors declare that they have no conflict of interest.
**Acknowledgement**
Resources supporting this work were provided by the NASA High-End Computing (HEC)
Program through the NASA Advanced Supercomputing (NAS) division at Ames Research Center.
We acknowledge the funding support from NASA OCO-2/3 Science Team, Carbon Monitoring
System (CMS), and Making Earth Science Data Records for Use in Research Environments
(MEaSUREs) programs. Tomohiro Oda is supported by the NASA Carbon Cycle Science program
(grant no. NNX14AM76G). We acknowledge EU and NERC (UK) funding to Emanuel Gloor,
University of Leeds which substantially contributed to the INPE Amazon greenhouse sampling
program. CarbonTracker Europe results provided by Wageningen University in collaboration with
the ObsPack partners (http://www.carbontracker.eu). Part of the research was carried out at Jet
Propulsion Laboratory, Caltech.

**Appendix A**
As shown in Kalnay (2003):
$RMSE^2 = \frac{1}{n}\sum_{i=1}^{n}(R_{i,i} + (HP^aH^T)_{i,i})$    (A.1)
where $R_{i,i}$ is the $i^{th}$ aircraft observation error variance, and $P^a$ is the posterior flux error covariance.
The $H$ is linearized observation operator, which transfers posterior flux errors to aircraft
observation space, and $H^T$ is its adjoint. In the Monte Carlo method, the posterior flux error
covariance $P^a$ is approximated by:
$P^a = \frac{1}{nens}X^aX^{aT}$ (A.2)
where $X^a$ is the ensemble perturbations written as:
$X^a = x^a - \bar{x}^a$ (A.3)
where $x^a$ is the ensemble posterior fluxes from Monte Carlo, and $\bar{x}^a$ is the mean.
Therefore, $HP^aH^T$ can be written as:
$HP^aH^T = \frac{1}{nens}[h(x^a) - h(\bar{x}^a)][h(x^a) - h(\bar{x}^a)]^T$ (A.4)
The sum of diagonal elements in the right-hand side of A.4 is the same as the definition of $RMSE_{MC}$
in the main text.
Therefore, when the posterior flux uncertainty estimated by Monte Carlo method represents the
actual uncertainty in posterior fluxes, equation (A.1) can be written as:
$RMSE^2 = \frac{1}{n}\sum_{i=1}^{n}R_{i,i} + RMSE_{MC}^2$    (A.5).
It is the same as equation (4) in the main text.
**Appendix B**
In this Appendix, we include figures to support the main text.

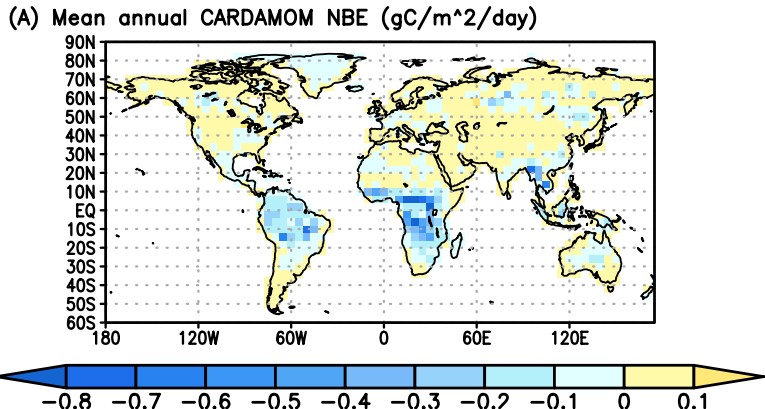

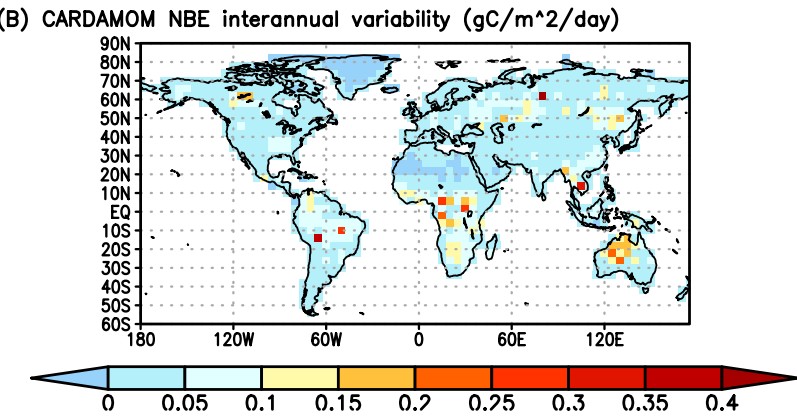


**Figure B1 Annual mean net biosphere exchanges from CARDAMOM (A) and its interannual**
**variability between 2010 and 2017 (B).**

(A) CARDAMOM absolute NBE (gC/m^2/day) in July 2010

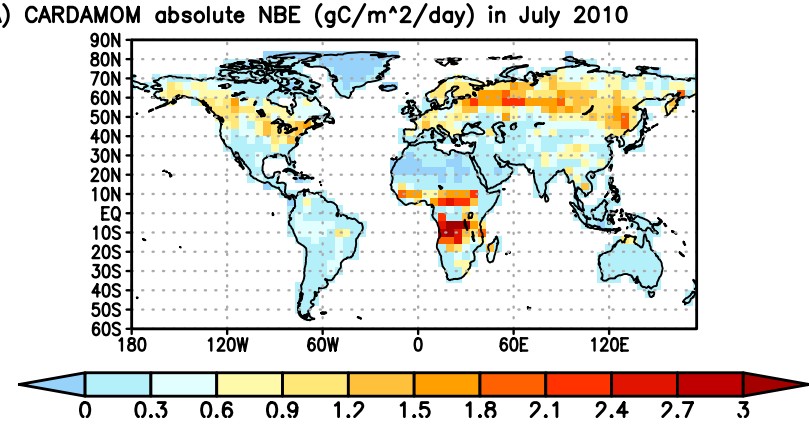

(B) CARDAMOM NBE uncertainty in July 2010 (gC/m^2/day)

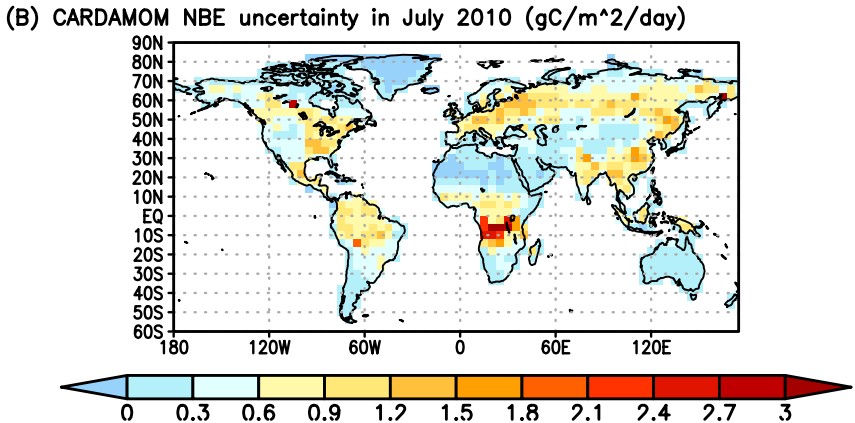


**Figure B2 An example of absolute mean NBE (A) and its uncertainty (B) simulated by CARDAMOM. This is for July 2010.**


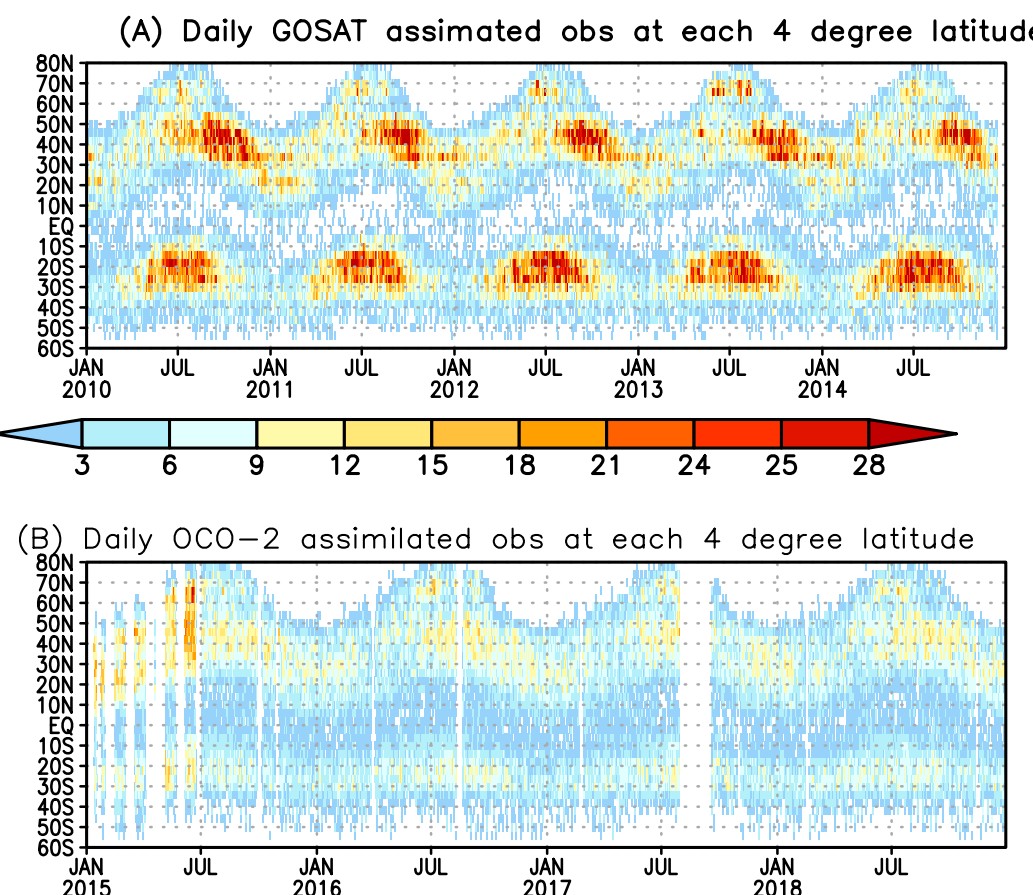

Figure B3 Daily number of ACOS-GOSAT b7.3 (A) and OCO-2 super observations (B) assimilated in the top-down inversions.

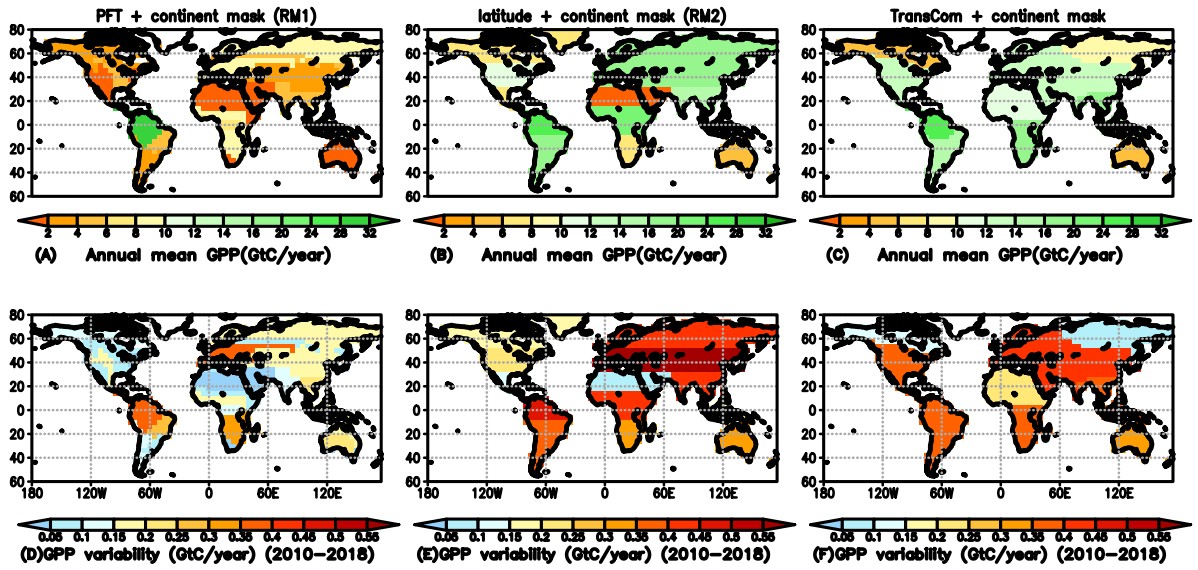

Figure B4 Regional mean FlUXSAT GPP and its variability between 2010 −2018. (A, B, and C)
Regional mean GPP aggregated with the three regional masks; (D, E, and F) GPP variability
between 2010 −2018.  Unit: GtC/year.


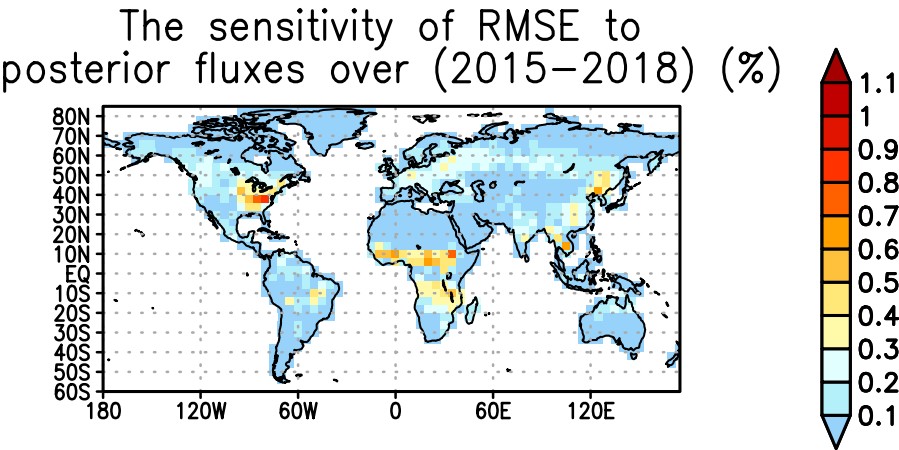

Figure B5 The relative sensitivity of root mean square errors (RMSE) of posterior $CO_2$ (Figure 9
in the main text) relative to NBE at every grid point. The adjoint model is carried out over Sep
2014–Dec 2018.


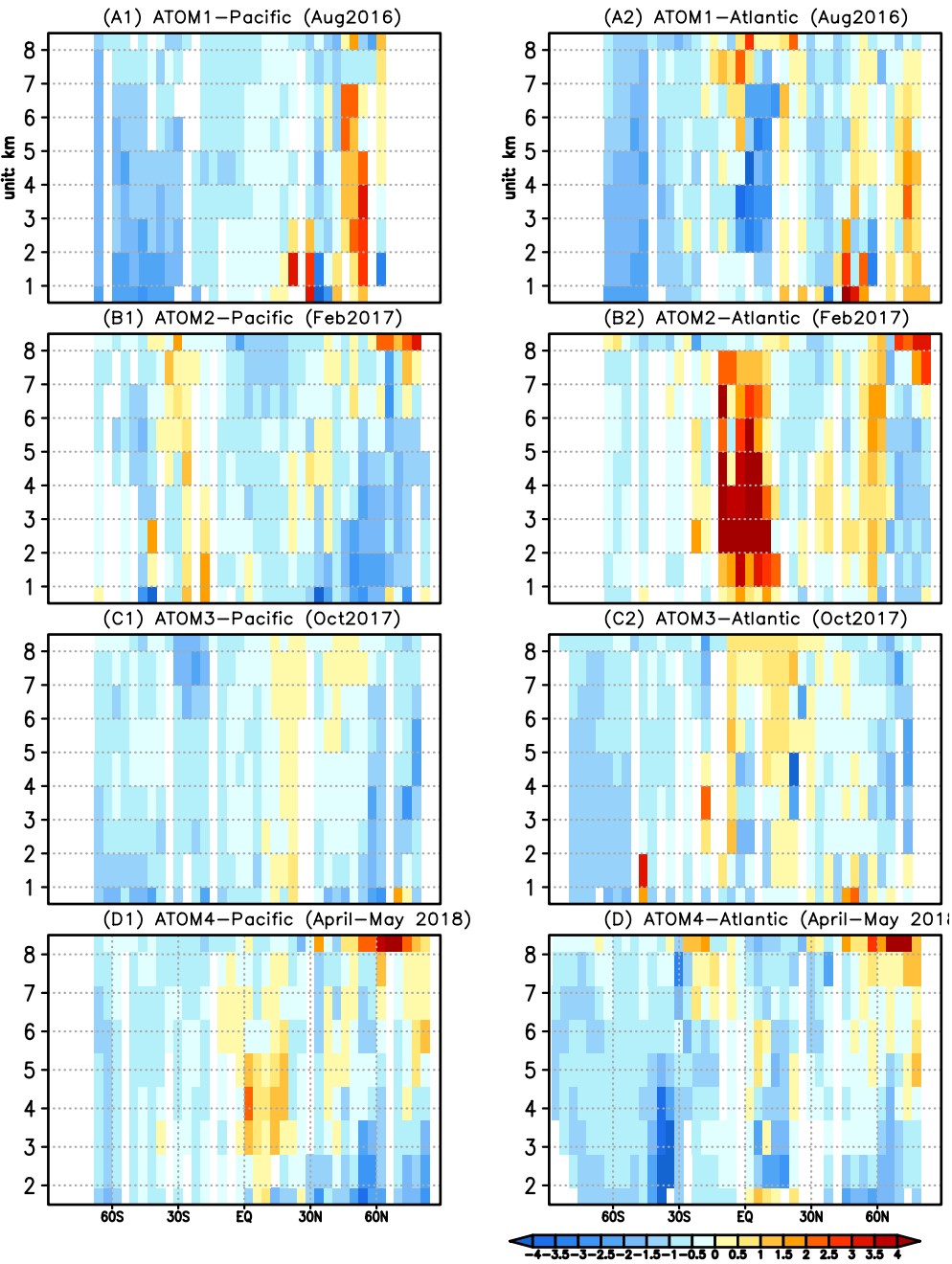

Figure B6 Differences between posterior $CO_2$ and ATOM 1-4 aircraft $CO_2$ observations over the
Pacific (A1-D1) and Atlantic Ocean (A2-D2) as a function of latitude and altitude (unit: km).
Unit: ppm.

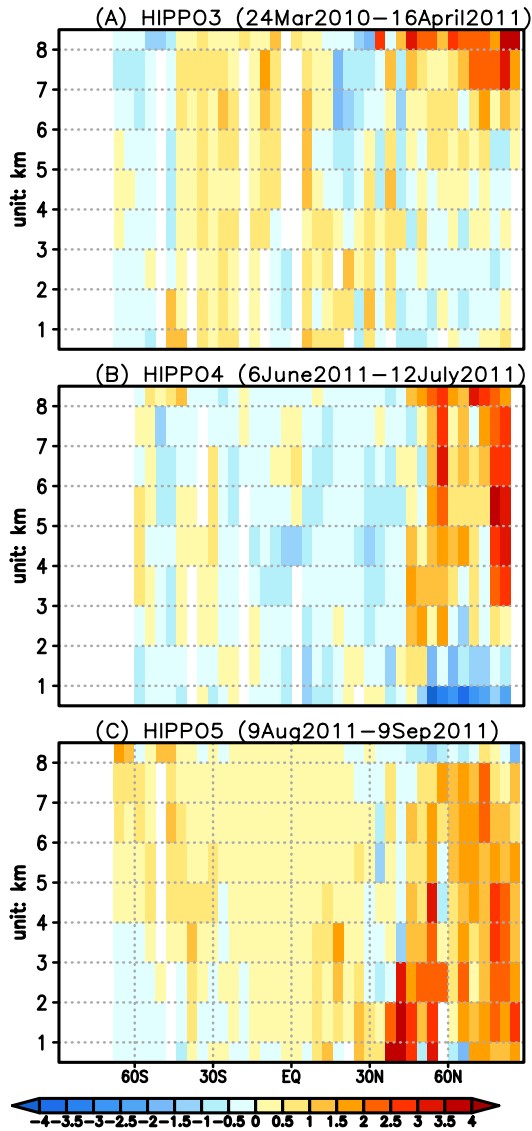

Figure B7 Differences between posterior $CO_2$ and HIPPO 3-5 aircraft $CO_2$ observations over the
Pacific (A-C) as a function of latitude and altitude. Unit: ppm.

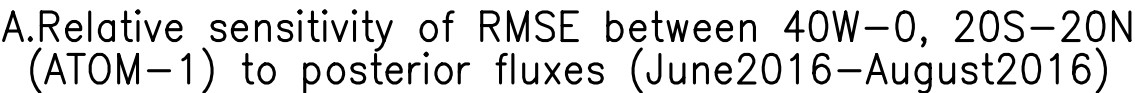

## A.Relative sensitivity of RMSE between 40W−0, 20S−20N (ATOM−1) to posterior fluxes (June2016−August2016)

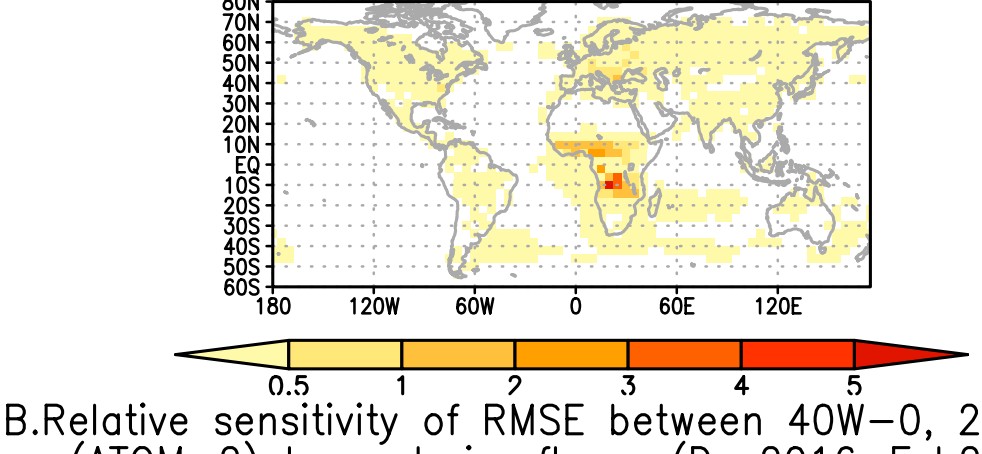

## B.Relative sensitivity of RMSE between 40W−0, 20S−20N (ATOM−2) to posterior fluxes (Dec2016−Feb2017)

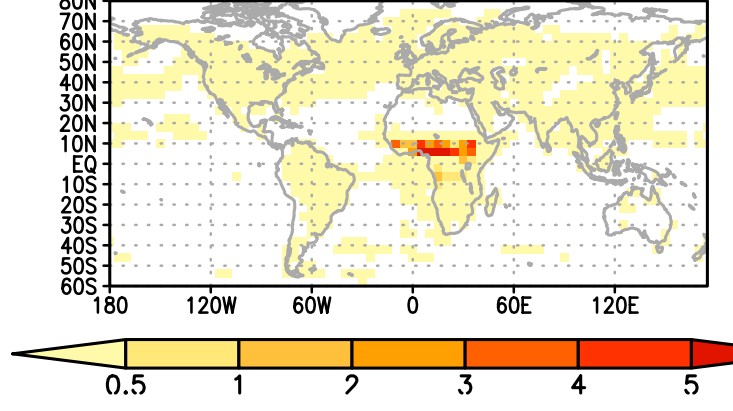

Figure B8 The relative sensitivity of RMSE of posterior $CO_2$ to NBE over land and air-sea net
carbon exchange over ocean at every grid point. The RMSE is calculated against aircraft $CO_2$
observations from ATom-1 (A) and ATom-2 (B) between 40°W-0°, 20°S-20°N. The adjoint
model is carried out over June – August 2016 (A) and Dec 2016 – Feb 2017 (B). Unit: %.




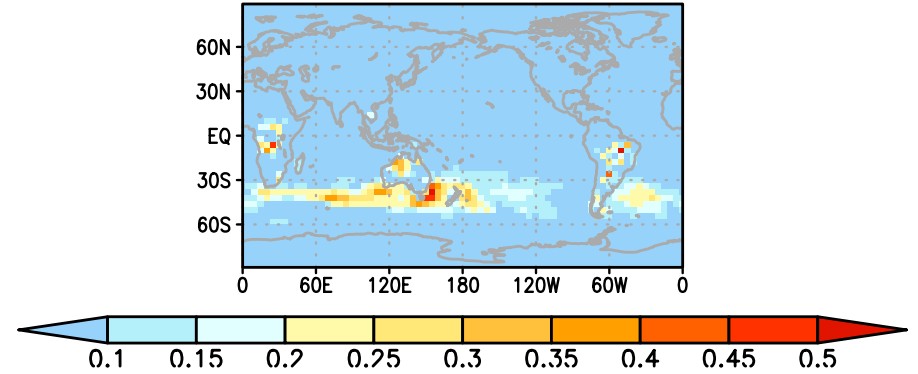



Figure B9 The relative sensitivity of RMSE of posterior to NBE over land and air-sea net carbon
exchange over ocean at every grid point. The RMSE is calculated against aircraft $CO_2$ observations
from ATom-1 between 175°W-20°W, 80°S-30°S. The adjoint model is carried out over June –
August 2016. Unit: %.





## Relative sensitivity of RMSE between 180W−130W, 50N−90N (HIPPO−4) to posterior fluxes (Apr−July)

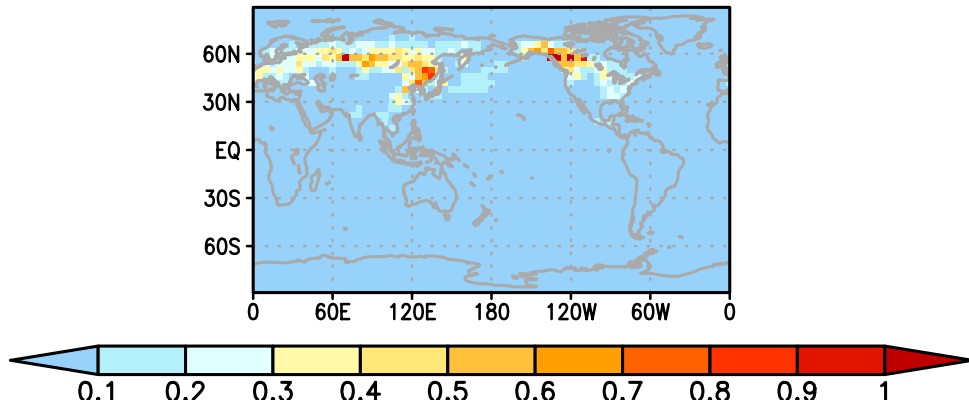


Figure B10 The relative sensitivity of RMSE of posterior to NBE over land and air-sea net carbon
exchange over ocean at every grid point. The RMSE is calculated against aircraft $CO_2$ observations
from HIPPO-4 between 180°W-130°W, 50°N-90°N. The adjoint model is carried out over April
– July 2011. Unit: %.

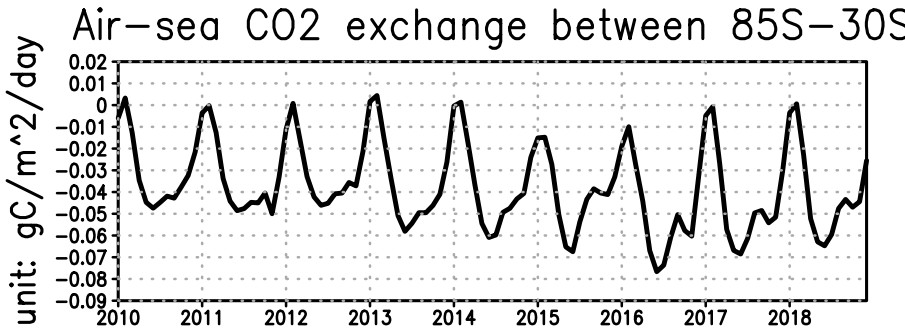

Figure B11 Monthly posterior air-sea $CO_2$ exchanges between 85°S-30°S. (unit: $gC/m^2/day$)

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

**NOAA MBL reference and posterior CO₂ before the comparison.**


**Table: 1 Configurations of the CMS-Flux atmospheric inversion system**

|  | **Model setup** | **Configuration** | **Reference** |
|---|---|---|---|
| **Inversion general setup** | Spatial scale | Global | -- |
|  | Spatial resolution | 4° latitude x 5° longitude |  |
|  | Time resolution | monthly |  |
|  | Minimizer of cost function | L-BFGS | Byrd et al., 1994; Zhu et al., 1997 |
|  | Control vector | Monthly net terrestrial biosphere fluxes and ocean fluxes |  |
| **Transport model** | Model name | GEOS-Chem and its adjoint | Suntharalingam et al., 2004 Nassar et al., 2010 Henze et al., 2007 |
|  | Meteorological forcing | GEOS-5 (2010–2014) and GEOS-FP (2015–2019) | Rienecker et al., 2008 |



**Table: 2 Description of the prior fluxes and assumed uncertainties in the inversion system**

| Prior fluxes | Terrestrial biosphere fluxes | Ocean fluxes | Fossil fuel emissions |
|---|---|---|---|
| Model name | CARDAMOM-v1 | ECCO-Darwin | ODIAC 2018 |
| Spatial resolution | 4° x 5° | 0.5° | 1° x 1° |
| Frequency | 3-hourly | 3-hourly | hourly |
| Uncertainty | Estimated from CARDAMOM | 100% same as Liu et al. (2017) | No uncertainty |
| References | Bloom et al., 2006; 2020 | Brix et al, 2015; Carroll et al., 2020 | Oda et al., 2016; 2018 |





**Table: 3 Description of observation and evaluation dataset. Data sources are listed in Table 7.**

| | Dataset name and version | References |
|---|---|---|
| Satellite $X_{CO2}$ | ACOS-GOSAT v7.3 | O'Dell et al., (2012) |
| | OCO-2 v9 | O'Dell et al., (2018) |
| Aircraft $CO_2$ observations | ObsPack OCO-2 MIP | CarbonTracker team (2019) |
| | HIPPO 3-5 | Wofsy et al. (2011) |
| | ATom 1-4 | Wofsy et al. (2018) |
| | INPE | Gatti et al., (2014) |
| | ORCAS | Stephens et al. (2017) |
| | ACT-America | Davis et al. (2018) |
| NOAA marine boundary layer (MBL) reference | NOAA MBL reference | Conway et al., 1994 |
| GPP | FLUXSAT-GPP | Joiner et al., (2018) |
| Top-down NBE estimates constrained by surface $CO_2$ | CarbonTracker-Europe | van der Laan-Luijkx et al. (2017)<br>Peters et al., (2010)<br>Peters et al. (2007) |
| | Jena CarbonScope s10oc_v2020 | Rödenbeck et al., 2003 |
| | CAMS v18r1 | Chevallier et al., 2005 |




**Table: 4 Latitude and longitude ranges for seven sub regions.**

| Region | Alaska | Mid-lat NA | Europe | East Asia | South Asia |
|---|---|---|---|---|---|
| **Longitude range** | 180°W–125° W | 125°W–65°W | 5°W–45°E | 110°E–160°E | 65°E–110°E |
| **Latitude range** | 58°N–89°N | 22°N-58°N | 30°N–66°N | 22°N–50°N | 10°S–32°N |
| **Region** | Africa | South America | Australia | Southern Ocean | |
| **Longitude range** | 5°W–55°E | 95°W–50°W | 120°E–160°E | 110°W–40°E | |
| **Latitude range** | 2°N–18°N | 20°S–2°N | 45°S–10°S | 80°S–30°S | |



**Table: 5 List of the data products.**

| Product | Spatial resolution | Temporal resolution when applicable | Data format | Sample data description in the text |
|---|---|---|---|---|
| **Total fossil fuel, ocean, and land fluxes** | Global | Annual | *csv* | Figure 4 (section 4.1) |
| **Climatology mean NBE, variability, and uncertainties** | PFT and continents based 28 regions | N/A | *csv* | Figure 5 (section 4.2) |
| | Geographic-based 13 regions | | *csv* | |
| | TransCom regions | | *csv* | |
| **Hemispheric NBE and uncertainties** | NH (20°N-90°N), tropics (20°S-20°N), and SH (60°S-20°S) | Annual | *csv* | Figure 6 (section 4.3) |
| **NBE variability and uncertainties** | PFT and continents based 28 regions | Annual | *csv* | Figure 7 (section 4.3) |
| | Geographic -based 13 regions | | *csv* | |
| | TransCom regions | | *csv* | |
| **NBE seasonality and its uncertainties** | PFT and continents based 28 regions | Monthly | *csv* | Figure 8 (section 4.4) |
| | Geographic -based 13 regions | | *csv* | |
| | TransCom regions | | *csv* | |
| **Monthly NBE and uncertainties** | PFT and continents based 28 regions | Monthly | *csv* | N/A |
| | Geographic -based 13 regions | | *csv* | |
| | TransCom | | *csv* | |
| **Gridded posterior NBE, air-sea carbon exchanges, and uncertainties** | 4° (latitude) x 5° (longitude) | Monthly | *NetCDF* | N/A |
| **Gridded prior NBE and air-sea carbon exchanges** | 4° (latitude) x 5° (longitude) | Monthly and 3-hourly | *NetCDF* | N/A |
| **Gridded fossil fuel emissions** | 4° (latitude) x 5° (longitude) | Monthly mean and hourly | *NetCDF* | N/A |
| **Region masks** | PFT and continents based 28 regions | N/A | csv | Figure 3 (section 2.4) |
| | Geographic -based 13 regions | | | |
| | TransCom regions | | | |



**Table: 6 The nine-year mean regional annual fluxes, uncertainties, and variability. Regions are based on the mask shown in Figure 5A (Figure 5.csv). Unit: GtC/year**

| Region name (Figure4.csv) | Mean NBE | Uncertainty | Variability |
|---|---|---|---|
| NA shrubland | -0.14 | 0.02 | 0.05 |
| NA needleleaf forest | -0.22 | 0.04 | 0.06 |
| NA deciduous forest | -0.2 | 0.04 | 0.07 |
| NA crop natural vegetation | -0.41 | 0.06 | 0.18 |
| NA grassland | -0.04 | 0.03 | 0.03 |
| NA savannah | 0.03 | 0.02 | 0.03 |
| Tropical South America (SA) evergreen broadleaf | 0.04 | 0.1 | 0.28 |
| SA savannah | -0.09 | 0.06 | 0.18 |
| SA cropland | -0.07 | 0.03 | 0.07 |
| SA shrubland | -0.03 | 0.02 | 0.08 |
| Eurasia shrubland savanna | -0.44 | 0.07 | 0.14 |
| Eurasia needleleaf forest | -0.41 | 0.07 | 0.12 |
| Europe cropland | -0.46 | 0.09 | 0.16 |
| Eurasia grassland | 0.02 | 0.08 | 0.13 |
| Asia cropland | -0.37 | 0.13 | 0.08 |
| India | 0.14 | 0.09 | 0.14 |
| Tropical Asia savanna | -0.12 | 0.11 | 0.08 |
| Tropical Asia evergreen broadleaf | -0.09 | 0.09 | 0.12 |
| Australia (Aus) savannah grassland | -0.11 | 0.02 | 0.09 |
| Aus shrubland | -0.07 | 0.01 | 0.05 |
| Aus cropland | -0.01 | 0.01 | 0.03 |
| African (Afr) northern shrubland | 0.04 | 0.02 | 0.03 |
| Afr grassland | 0.03 | 0.01 | 0.01 |
| Afr northern savanna | 0.54 | 0.15 | 0.49 |
| Afr southern savanna | -0.27 | 0.18 | 0.33 |
| Afr evergreen broadleaf | 0.1 | 0.07 | 0.09 |
| Afr southern shrubland | 0.01 | 0.01 | 0.01 |
| Afr desert | 0.06 | 0.01 | 0.04 |





**Table: 7 Lists of data sources used in producing and evaluating posterior NBE product.**

| Data name | Data Source |
|---|---|
| ECCO-Darwin ocean fluxes | https://doi.org/10.25966/4v02-c391 |
| CARDAMOM NBE and uncertainties | https://doi.org/10.25966/4v02-c391 |
| ODIAC | http://db.cger.nies.go.jp/dataset/ODIAC/DL_odiac2019.html |
| GOSAT b7.3 | https://oco2.gesdisc.eosdis.nasa.gov/data/GOSAT_TANSO_Level2/ACOS_L2S.7.3/ |
| OCO-2 b9 | https://disc.gsfc.nasa.gov/datasets?page=1&keywords=OCO-2 |
| ObsPack | https://www.esrl.noaa.gov/gmd/ccgg/obspack/data.php |
| ATom 1-4 | https://daac.ornl.gov/ATOM/guides/ATom_merge.html |
| HIPPO 3-5 | https://www.eol.ucar.edu/field_projects/hippo |
| INPE | https://www.esrl.noaa.gov/gmd/ccgg/obspack/data.php?id=obspack_co2_1_INPE_RESTRICTED_v2.0_2018-11-13 and |
| FLUXSAT-GPP | https://gs614-avdc1-pz.gsfc.nasa.gov/pub/tmp/FluxSat_GPP/ |
| NOAA MBL reference | https://www.esrl.noaa.gov/gmd/ccgg/mbl/index.html |
| CarbonTracker-Europe NBE | https://www.carbontracker.eu/download.shtml |
| Jena CarbonScope NBE | http://www.bgc-jena.mpg.de/CarboScope/?ID=s |
| CAMS NBE | https://apps.ecmwf.int/datasets/data/cams-ghg-inversions/?date_month_slider=2009-12,2018-12¶m=co2&datatype=ra&version=v17r1&frequency=mm&quantity=surface_flux |
| Posterior NBE | https://doi.org/10.25966/4v02-c391 |


