# Peer review of "Carbon Monitoring System Flux Net Biosphere Exchange 2020 (CMS-Flux NBE 2020)"

_Earth System Science Data, 2020_

## Referee Comment (RC1) · Julia Marshall (Referee) · 28 Jul 2020

This article documents the land biosphere prior and the optimized land biosphere and ocean posterior fluxes resulting from a 2010-2018 inversion using the Carbon Monitoring System (CMS) modelling system. The observational inputs are restricted to satellite measurements of XCO2, based on GOSAT for 2010-2014 and OCO-2 from 2015-2018.

My first hesitation is based on the appropriateness of publishing optimized fluxes in ESSD. These are not measurements, but rather, by definition, a model-based interpretation of measurements. In the description of the Aims and Scope of the journal it states that: "Any interpretation of data is outside the scope of regular articles." On some level

this is a philosophical distinction, and it seems this article has made it through the first quick review process, so I have to assume that the editor does not see a major problem here.

My next hesitation is on the criterion of "Completeness". This is one of the categories reviewers are asked to assess, stating that: "A data set or collection must not be split intentionally, for example, to increase the possible number of publications. It should contain all data that can be reviewed without unnecessary increase of workload and can be reused in another context by a reader."

The current paper seems to be a classic case of withholding part of the dataset to release it in a future publication. The modelling system used has been well-documented in its ability to (quoting from the Introduction): "resolve regional fluxes, and also disentangle net biosphere exchange (NBE) into constituent carbon fluxes including plant gross primary productivity (GPP) and biomass burning through solar-induced fluorescence and carbon monoxide proxies, respectively (Bowman 53 et al, 2017, Liu et al., 2017)." So is that what is reported here? No, they have decided to only report NBE, stating that" Subsequent papers will present the partitioning of the NBE into constituent gross fluxes." This seems like a clear infringement of the "Completeness" criterion. My recommendation would be to include the optimized GPP, biomass burning, and respiration fluxes in the same data release, and have an accompanying analysis paper (in another journal) that goes into the interpretation of the retrieved signals. One of the unique strengths of the CMS is its partitioning of the net land biosphere fluxes, which most modellers do not claim to be able to do with any confidence. It is this partitioning that would make the resultant fluxes more interesting for comparison against other approaches.

Finally, my third hesitation is related to the data quality. This is not related to anything that the authors themselves have done wrong, but there are (still) clear limitations to using only satellite data in an inversion. This has been well documented in the literature (e.g. Basu et al., 2013; Chevallier et al., 2014), and leads to an unexpectedly
high source in northern Africa (and perhaps a too-large sink in Europe) that is hard to reconcile with bottom-up fluxes and inversions based on surface/in-situ measurements. The magnitude of this potential bias has decreased in more recent retrieval versions but has not disappeared.

This discrepancy is clearly seen in the limited validation that is presented here, when the optimized concentrations are compared to the Atlantic ATom-1 and -2 flights. At least this inconsistency with independent data is documented: hopefully potential users of this dataset will not assume that their model is wrong simply because it disagrees with the CMS fluxes. One potential improvement here would be to include also fluxes optimized based on surface-based measurements, to give some idea of the uncertainty in the fluxes as a result of the choice of input data. (The estimated fluxes will most likely not agree within the stated uncertainties.)

This is downplayed by comparing the global land biosphere budget to widely accepted values from the Global Carbon Project (Friedlingstein et al., 2019), rather than the regional breakdown. Comparing to Figure 8 of Friedlingstein et al. (2019) it seems that the tropics are a more substantial source and the extratropics a more substantial sink than is seen within the spread of inverse models included in the GCP analysis.

Another potential limitation to the usefulness of the data is the underwhelming resolution. Monthly fluxes at 4 x 5 degree resolution are no longer really state-of-the-art. One of the arguments for using satellite measurements is the higher spatial resolution of the signals that can be resolved compared to the rather sparse surface-based network. This dataset is not exploiting to this strength.

Based on these concerns, I would not recommend publishing this paper in ESSD in its current form.

Other comments:

Regarding the completeness of the dataset presented, I had some minor concerns.

I tried to check the availability of the datasets linked to here, and found that I was not sure which version of the ECCO-Darwin fluxes had been used: the data portal lists several different options. I was not even entirely sure if the ocean fluxes had been optimized, but the netCDF gridded fluxes describe the ocean fluxes as "posterior ocean fluxes,2010-2014 constrained by GOSAT, 2015-2018 constrained by OCO2", so I assume that they are not identical to the prior. In any case, this could be clarified. Similarly, the paper mentions that FLUXCOM-GPP is one of the inputs to CARDAMOM, but there is more than one version of this product as well. Even CARDAMOM comes in different flavours, I believe, based on the documentation in the cited papers. For completeness it would be suitable to include all the prior and posterior fluxes in the dataset - including the anthropogenic fluxes which are not optimised. Only then can the full budget be assessed.

I do not see the purpose of providing the monthly fluxes (with uncertainty) at 13 different FLUXNET sites. As far as I can tell, these are extracted directly from the model, and do not represent additional downscaling or enhanced temporal resolution. The benefit of this (and the rationale for the selection of these specific sites) is not clear to me. The measured monthly mean NBE at these sites is not included for comparison, nor is any validation using FLUXNET sites provided in the manuscript. It seems redundant.

I was surprised by the choice of the masks used for aggregation of fluxes. If two masks are included, why not include the broadly-applied TransCom mask? The benefit of such a common mask is the ease of comparison. Yes, a user may apply his or her own mask to the data, but it really does not add much in terms of space (22 regions with monthly resolution), and would facilitate comparison with already available model output. This is likely of more general application than the two custom masks given here.

Minor/typographical comments:

L29: "from Greenhouse" -> "from the Greenhouse"

L30: remove "the" before NASA

L49: Crowell et al. 2019 is an odd choice as an example of inversions based on surface CO2 observations, as this was explicitly not the focus of the publication.

L55: The NBE are far -> NBE is far

L108: suggest " North America (NA)" -> "North American" (abbreviation is established elsewhere, and adjectival form fits better here)

L122: Section 8 is -> Section 8 describes the

L128: "that no" -> "no"

L151: its -> their

L161: from2010 (missing space)

L169: CARDAMON -> CARDAMOM

L184: by ACOS -> by the ACOS

L185: maximize -> maximizes

L193: land nadir good quality observations -> land nadir observations flagged as being of good quality

L221: of OCO-2 -> of the OCO-2

L272: over Pacific, and -> over the Pacific, but

L277: its -> their

L279: each nine -> each of nine

L283: fractions sampled at ith aircraft locations -> fraction sampled at the ith aircraft location

L285: of mean -> of the mean

L287: either posterior fluxes or transport -> either the posterior fluxes or the transport

[Figure]

L288: posterior fluxes -> the posterior fluxes (twice)

L315: of RMSE to posterior flux using GEOS-Chem -> of the RMSE to the posterior flux using the GEOS-Chem
 L342: Please rewrite the first sentence.

L355 & L366: by NOAA -> by the NOAA

L360: of posterior -> of the posterior

L371: calculate -> calculated

L375: with GCP -> with the GCP (or, "with the range estimated by the GCP,")

L382: shows large -> shows that large

L382: Southern -> the Southern

L383: eastern -> the eastern

L409: or weakly -> or are weakly

L410: during 2015 -> during the 2015

L415: Pouter -> Poulter

L416: capitalisation weird in "tropical south America Savanna"
 L440: above planetary -> above the planetary

L446: used NOAA -> used the NOAA

L448: is equal or above -> is greater than or equal to
 L450 & L482 & L497: Southern Ocean -> the Southern Ocean (as an aside: the weaker seasonality certainly plays a role, but this was also a "problem region" in comparison to Atom-1 measurements, so perhaps there is something else going on there...)

Figure 2: It is really difficult to tell the regions apart on this map. The blues in North America, for instance, really blend together.

Figure 9: This figure seems unnecessarily cramped. Perhaps split it into two? For instance, I could barely tell that the bars were really blue without zooming way in, as they are so tiny.

General figure comment: Something is a bit off with the rendering of the digits in your colour bars, making the bottom bar of "2"s disappear and making a gap in the bottom of round digits.

L466: from Monte -> from the Monte

L478: What is meant here? "either transport or low of posterior flux uncertainty estimates": Perhaps, "either transport errors or too low values for posterior flux uncertainties"?

L480: of flight -> of the flight

L493: defined -> as defined

L496: These -> The, also, specify which ratio you mean "between RMSE and RMSEMC"

L500: Pacific -> the Pacific

L503: these -> the

L504: with 4° x 5° resolution transport model -> using a transport model with only 4° x 5° resolution

L513-515: A bit awkward, please rephrase.

L516: indicates small -> indicates a small

L518: of posterior -> of the posterior

L533: to FLUXSAT -> to the FLUXSAT

L541: "needs caution" -> perhaps better: "calls for caution"?

L556: by GCP -> by the GCP

L562: atmospheric -> the atmospheric

L563: level -> levels

L576: provide support the monitoring of the regional contributions to the changes inÂăatmospheric. . . I'm not sure about this, perhaps: "support the monitoring of the regional (biospheric) contributions to changes in atmospheric. . ."?

L580 & L582: data is -> data are

L582: ensemble posterior -> ensemble of posterior
* * *

---

## Referee Comment (RC2) · Anonymous Referee #2 · 7 Aug 2020

General comments.

The authors present a brief description and evaluation results for the inverse model estimated $CO2$ fluxes for 2010-2018, based on observations by GOSAT and OCO-2 satellites. The data presented in the dataset are produced with the same model that was applied in several research papers and have mostly been used for estimating the variability and anomalies in the global carbon cycle at the regional and global scale. The satellite-based flux inversions proved to be useful in constraining large regional scale response of the natural carbon cycle to climate anomalies, droughts, heatwaves, such as those driven by the El-Nino cycle. In this context, the presented data can become a useful asset for those studying the carbon cycle variability at regional scale and its connection to the climate anomalies. On the negative side, there are desirable components in the evaluation, such as analysis of the $CO_2$ flux seasonal cycle, its comparison with inverse model estimates made with ground-based observations, or other independent estimates, such as based on flux tower data. The same can be said on comparison with observed $CO_2$ concentration at background monitoring sites, such as the NOAA flask sampling network. In case there are identified biases in such comparison, it would be possible to advise the users to restrict the use of the data to studying the flux anomalies rather than using the fluxes for forward simulations, comparing with surface fluxes and using in ecosystem model optimization, where seasonal cycle performance is important. The authors should clearly state such limitations so that the users can have enough information on how to make best use of the provided data. The paper is well written and can be accepted after minor revision addressing the comments and suggestions.

Detailed comments.

Notable deficiency: NBE flux evaluation looks somewhat qualitative. Based on data presented in the paper, and data provided on the data distribution site it is difficult to compare the NBE fluxes to alternative estimates. The 28-region data is provided, but it doesn't look directly mappable to widely used Transcom-3 22 region map. Recommend adding comparison figure (similar to Figure 8) of the seasonal flux climatology on Transcom3 22 regions or the authors-proposed 28 regions to other available estimates such as CAMS inversion fluxes (based on Chevallier et al. 2010) or FLUXCOM fluxes (Jung et al. 2020)

Line 208 It looks like presented bias figures (below 0.1 ppm) are related to global mean bias, are the bias values available as seasonal mean values by latitude or TCCON site? Are retrieved and bias-corrected concentrations consistent with model simulations optimized with ground-based observations?

Line 1031 Figure 8. Although the seasonally varying fluxes look to be in a reasonable range it is very much advisable to compare/plot along with observed or observationbased fluxes, such as FLUXCOM NEE product (Jung et al. 2020).

Technical corrections.

Line 191 Looks anomalous, to have 2000 good quality retrievals available on a single day in the ACOS-GOSAT dataset (appears significantly larger than average).

Line 193 Need to state how good quality is defined (what value of the quality flag is used)?

Line 199 Is 'super observations' a good term to name 100 km (~12 sec) average data?

Line 228 The statement "For large-order systems, the posterior errors cannot be explicitly calculated" can be argued. Posterior flux uncertainty projected to regions can be estimated analytically using recipes provided by (Fisher and Courtier, 1995) or (Meirink et al, 2008), using either flux singular vectors or flux increments obtained on course of the iterative optimization (eg Niwa and Fujii, 2020). Using random perturbations is simpler and is used widely, but that doesn't mean that the more accurate method is impossible to apply.

Line 240 Common perception is that tower footprint size is less than 1 km, based on estimates by Baldocchi, (1997) and others. The citation by Running et al (1999) of 'several km2' may refer to the upper range. They (Running et al 1999) also consider 1–3 km2 and 1 km2 as typical values throughout their paper.

References

Baldocchi, D.: Flux Footprints Within and Over Forest Canopies, Bound.-Lay. Meteorol., 85, 273–292, 1997.

Chevallier, F., et al.: CO2 surface fluxes at grid point scale estimated from a global 21-year reanalysis of atmospheric measurements, J. Geophys. Res., 115, D21307, doi:10.1029/2010JD013887, 2010.

Fisher, M., and Courtier, P.: Estimating the covariance matrices of analysis and forecast

error in variational data assimilation, ECMWF Technical memo. 220, Shinfield Park, Reading, 26, 1995.

Jung, M., Schwalm, C., Migliavacca, M., Walther, S., et al: Scaling carbon fluxes from eddy covariance sites to globe: synthesis and evaluation of the FLUXCOM approach, Biogeosciences, 17, 1343–1365, https://doi.org/10.5194/bg-17-1343-2020, 2020.

Meirink, J., Bergamaschi, P., and Krol, M.: Four-dimensional variational data assimilation for inverse modelling of atmospheric methane emissions: method and comparison with synthesis inversion, Atmospheric Chemistry and Physics, 8, 6341-6353, 10.5194/acp-8-6341-2008, 2008.

---

## Author Comment (AC2) · 2 Sep 2020

Thanks for the constructive comments. Please see our response below. Anonymous Referee #2 Original: The authors present a brief description and evaluation results for the inverse model estimated CO2 fluxes for 2010-2018, based on observations by GOSAT and OCO-2 satellites. The data presented in the dataset are produced with the same model that was applied in several research papers and have mostly been used for estimating the variability and anomalies in the global carbon cycle at the regional and global scale. The satellite-based flux inversions proved to be useful in constraining large regional scale response of the natural carbon cycle to climate anomalies, droughts, heatwaves, such as those driven by the El-Nino cycle. In this context, the presented data can become a useful asset for those studying the carbon cycle variability at regional scale and its connection to the climate anomalies. On the negative side, there are desirable components in the evaluation, such as analysis of the $CO_2$ flux seasonal cycle, its comparison with inverse model estimates made with ground-based observations, or other independent estimates, such as based on flux tower data. The same can be said on comparison with observed $CO_2$ concentration at background monitoring sites, such as the NOAA flask sampling network. In case there are identified biases in such comparison, it would be possible to advise the users to restrict the use of the data to studying the flux anomalies rather than using the fluxes for forward simulations, comparing with surface fluxes and using in ecosystem model optimization, where seasonal cycle performance is important. The authors should clearly state such limitations so that the users can have enough information on how to make best use of the provided data. The paper is well written and can be accepted after minor revision addressing the comments and suggestions.

Response: We appreciate the constructive comments. In the revision, we will add comparison to NOAA background monitoring sites in terms of seasonal cycle. Please see our detailed response below.

Detailed comments. Original: Notable deficiency: NBE flux evaluation looks somewhat qualitative. Based on data presented in the paper, and data provided on the data distribution site it is difficult to compare the NBE fluxes to alternative estimates. The 28-region data is provided, but it doesn't look directly mappable to widely used Transcom-3 22 region map. Recommend adding comparison figure (similar to Figure 8) of the seasonal flux climatology on Transcom3 22 regions or the authors-proposed 28 regions to other available estimates such as CAMS inversion fluxes (based on Chevallier et al. 2010) or FLUXCOM fluxes (Jung et al. 2020).

Response: We will add the regional fluxes based on Transcom-3 22-region map in the revision.

Direct evaluation of regional NBE from top-down inversion is difficult, since there is

no direct regional NBE measurements available. In addition to the comparison with independent aircraft observations, which is commonly used in the inversion community, we have carried out two additional steps to link the posterior CO2 errors with underlying posterior fluxes. First, we have quantified the contributions of fluxes at each grid point to posterior CO2 errors, which identify regions that significantly contribute to the posterior CO2 errors. (Figure S8-S10). Second, we have evaluated the magnitude of posterior error estimates from Monte Carlo method using independent aircraft observations as described in 2.5.2 and Figures 9-11.

In the revision, we will add the comparison with NOAA background monitoring sites to identify any possible deficiency in the CO2 flux seasonal cycle and north-south gradient.

We will also add a comparison figure in the revision comparing to CAMS inversion or FLUXCOM fluxes.

Original: Line 208 It looks like presented bias figures (below 0.1 ppm) are related to global mean bias, are the bias values available as seasonal mean values by latitude or TCCON site? Are retrieved and bias-corrected concentrations consistent with model simulations optimized with ground-based observations? Response: O'Dell et al., (2018) evaluated OCO-2 retrievals at TCCON sites and compared the OCO2 with model simulations optimized with ground-based observations. We do not evaluate data bias but refer to O'Dell et al, 2018 for bias characterization and mitigation at seasonal and latitudinal scales In the revision, we will cite the numbers from O'Dell et al. (2018).

O'Dell, C. W., Eldering, A., Wennberg, P. O., Crisp, D., Gunson, M. R., Fisher, B., Frankenberg, C., Kiel, M., Lindqvist, H., Mandrake, L., Merrelli, A., Natraj, V., Nelson, R. R., Osterman, G. B., Payne, V. H., Taylor, T. E., Wunch, D., Drouin, B. J., Oyafuso, F., Chang, A., McDuffie, J., Smyth, M., Baker, D. F., Basu, S., Chevallier, F., Crowell, S. M. R., Feng, L., Palmer, P. I., Dubey, M., García, O. E., Griffith, D. W. T., Hase, F., Iraci, L. T., Kivi, R., Morino, I., Notholt, J., Ohyama, H., Petri, C., Roehl, C. M., Sha, M. K.,

Strong, K., Sussmann, R., Te, Y., Uchino, O., and Velazco, V. A.: Improved retrievals of carbon dioxide from Orbiting Carbon Observatory-2 with the version 8 ACOS algorithm, Atmos. Meas. Tech., 11, 6539–6576, https://doi.org/10.5194/amt-11-6539-2018, 2018.

Original: Line 1031 Figure 8. Although the seasonally varying fluxes look to be in a reasonable range it is very much advisable to compare/plot along with observed or observation-based fluxes, such as FLUXCOM NEE product (Jung et al. 2020).

Response: We will comparison to observation-based fluxes in the revision.

Original: Line 191 Looks anomalous, to have 2000 good quality retrievals available on a single day in the ACOS-GOSAT dataset (appears significantly larger than average).

Response: it is a mistake. It should be monthly.

Original: Line 193 Need to state how good quality is defined (what value of the quality flag is used)?

Response: We will clarify how the good quality is defined.

Original: Line 199 Is 'super observations' a good term to name 100 km (_12 sec) average data?

Response: This term is from numerical weather prediction. We will cite relevant references in the revision.

Original: Line 228 The statement "For large-order systems, the posterior errors cannot be explicitly calculated" can be argued. Posterior flux uncertainty projected to regions can be estimated analytically using recipes provided by (Fisher and Courtier, 1995) or (Meirink et al, 2008), using either flux singular vectors or flux increments obtained on course of the iterative optimization (eg Niwa and Fujii, 2020). Using random perturbations is simpler and is used widely, but that doesn't mean that the more accurate method is impossible to apply.

Response: We will incorporate these comments in the revision.

Original: Line 240 Common perception is that tower footprint size is less than 1 km, based on estimates by Baldocchi, (1997) and others. The citation by Running et al (1999) of 'several km2' may refer to the upper range. They (Running et al 1999) also consider 1–3 km2 and 1 km2 as typical values throughout their paper.

Response: We will correct it.

————————————————————

---

## Author Response (AR1)

Dear Dr. Marshall,
We appreciate very much your comments. Please see our responses below.

*This article documents the land biosphere prior and the optimized land biosphere and ocean posterior fluxes resulting from a 2010-2018 inversion using the Carbon Monitoring System (CMS) modelling system. The observational inputs are restricted to satellite measurements of XCO2, based on GOSAT for 2010-2014 and OCO-2 from 2015-2018.*

***Original:*** *My first hesitation is based on the appropriateness of publishing optimized fluxes in ESSD. These are not measurements, but rather, by definition, a model-based interpretation of measurements. In the description of the Aims and Scope of the journal it states that: "Any interpretation of data is outside the scope of regular articles." On some level this is a philosophical distinction, and it seems this article has made it through the first quick review process, so I have to assume that the editor does not see a major problem here.*

**Response**: This is a reanalysis dataset, not a pure model simulation. It is a combination of observations and an *apriori* based on their respective error statistics. Reanalysis products have been used more broadly in research than the raw observations. For example, the meteorology reanalysis data sets (e.g., MERRA and ERA-5) have many more users than weather station data or satellite radiances.

***Original***: *My next hesitation is on the criterion of "Completeness". This is one of the categories reviewers are asked to assess, stating that: "A data set or collection must not be split intentionally, for example, to increase the possible number of publications. It should contain all data that can be reviewed without unnecessary increase of workload and can be reused in another context by a reader." The current paper seems to be a classic case of withholding part of the dataset to release it in a future publication. The modelling system used has been well-documented in its ability to (quoting from the Introduction): "resolve regional fluxes, and also disentangle net biosphere exchange (NBE) into constituent carbon fluxes including plant gross primary productivity (GPP) and biomass burning through solar-induced fluorescence and carbon monoxide proxies, respectively (Bowman et al, 2017, Liu et al., 2017)." So is that what is reported here? No, they have decided to only report NBE, stating that" Subsequent papers will present the partitioning of the NBE into constituent gross fluxes." This seems like a clear infringement of the "Completeness" criterion. My recommendation would be to include the optimized GPP, biomass burning, and respiration fluxes in the same data release, and have an accompanying analysis paper (in another journal) that goes into the interpretation of the retrieved signals. One of the unique strengths of the CMS is its partitioning of the net land biosphere fluxes, which most modellers do not claim to be able to do with any confidence. It is this partitioning that would make the resultant fluxes more interesting for comparison against other approaches.*

**Response**: We respectfully disagree that the data are incomplete per ESSD guidance. The NBE can be reviewed independently of component fluxes and can be re-used in many applications by the reader, e.g., comparison to DGVM output, understanding the carbon-climate feedbacks etc. We agree that NBE can be more richly understood by partitioning it into component fluxes. However, there are multiple ways that this can be done, whether through independent data streams or additional data streams (e.g., Liu et al., 2017; Bowman et al., 2017) or through a more sophisticated land-surface assimilation, e.g., Quetin et al, 2020. We do not want to prejudge that methodology or the additional data that might be used.

With respect to NBE, we report all the elements from the inversion system, including gridded fluxes, uncertainties, and regionally aggregated fluxes. We evaluate both the mean fluxes and the uncertainty estimates with independent observations.

We will modify the introduction to reflect the fact that we have not withheld relevant datasets.

G. R. Quetin, A. A. Bloom, K. W. Bowman, A. G. Konings, Carbon Flux Variability From a Relatively Simple Ecosystem Model With Assimilated Data Is Consistent With Terrestrial Biosphere Model Estimates. *J Adv Model Earth Sy*. **12** (2020), doi:10.1029/2019ms001889.

***Original***: *Finally, my third hesitation is related to the data quality. This is not related to anything that the authors themselves have done wrong, but there are (still) clear limitations to using only satellite data in an inversion. This has been well documented in the literature (e.g. Basu et al., 2013; Chevallier et al., 2014), and leads to an unexpectedly high source in northern Africa (and perhaps a too-large sink in Europe) that is hard to reconcile with bottom-up fluxes and inversions based on surface/in-situ measurements. The magnitude of this potential bias has decreased in more recent retrieval versions but has not disappeared. This discrepancy is clearly seen in the limited validation that is presented here, when the optimized concentrations are compared to the Atlantic ATom-1 and -2 flights. At least this inconsistency with independent data is documented: hopefully potential users of this dataset will not assume that their model is wrong simply because it disagrees with the CMS fluxes. One potential improvement here would be to include also fluxes optimized based on surface-based measurements, to give some idea of the uncertainty in the fluxes as a result of the choice of input data. (The estimated fluxes will most likely not agree within the stated uncertainties.) This is downplayed by comparing the global land biosphere budget to widely accepted values from the Global Carbon Project (Friedlingstein et al., 2019), rather than the regional breakdown. Comparing to Figure 8 of Friedlingstein et al. (2019) it seems that the tropics are a more substantial source and the extratropics a more substantial sink than is seen within the spread of inverse models included in the GCP analysis.*

**Response**: We agree with Dr. Marshall that the satellite-based NBE product is not perfect. However, neither is a surface-based inversion product nor is any assimilated product, e.g., ERA5. In particular, surface-based information used in the GCP analysis provides limited information on the tropics. Satellite-based NBE estimates have provided many new insights on the carbon cycle. For example, Basu et al. (2014) studied the flux seasonal variation over tropical Asia with top-down flux estimates based on GOSAT observations. Detmers et al. (2015) studied the 2011 anomalous carbon sink over Australia using NBE estimates based on GOSAT observations. Liu et al. (2018) CMS-Flux results showed excellent agreement with the North American carbon balance changes with in-situ approaches from Wolf et al. (2016). A snapshot of the differences between inversion systems has been documented in Crowell et al. (2019). Those differences will evolve even as a number of these systems converge on their inferences, (e.g., Gaubert et al., 2019).  Sharing the data with the broader community will accelerate its use in scientific exploration, and at the same time, will help identify possible deficiencies that further feeds back on future development.

In this paper, we evaluate the reported fluxes and corresponding uncertainties with independent aircraft observations using rigorous methodology. As noticed by Dr. Marshall, we also point to any possible deficiencies in the products based on these evaluations.

Basu, S., Krol, M., Butz, A., Clerbaux, C., Sawa, Y., Machida, T., Matsueda, H., Frankenberg, C., Hasekamp, O. P., and Aben, I. (2014),  The seasonal variation of the $CO_2$ flux over Tropical Asia estimated from GOSAT, CONTRAIL, and IASI, *Geophys. Res. Lett.*,  41,  1809– 1815, doi:10.1002/2013GL059105.

Detmers, R. G.,  Hasekamp, O.,  Aben, I.,  Houweling, S.,  van Leeuwen, T. T.,  Butz, A., Landgraf, J.,  Köhler, P.,  Guanter, L., and  Poulter, B. (2015),  Anomalous carbon uptake in Australia as seen by GOSAT, *Geophys. Res. Lett.*,  42,  8177– 8184, doi:10.1002/2015GL065161.

Crowell, S., Baker, D., Schuh, A., Basu, S., Jacobson, A. R., Chevallier, F., Liu, J., Deng, F., Feng, L., McKain, K., Chatterjee, A., Miller, J. B., Stephens, B. B., Eldering, A., Crisp, D., Schimel, D., Nassar, R., O'Dell, C. W., Oda, T., Sweeney, C., Palmer, P. I., and Jones, D. B. A.: The 2015–2016 carbon cycle as seen from OCO-2 and the global in situ network, Atmos. Chem. Phys., 19, 9797–9831, https://doi.org/10.5194/acp-19-9797-2019, 2019.

J. Liu, K. Bowman, N. C. Parazoo, A. A. Bloom, D. Wunch, Z. Jiang, K. R. Gurney, D. Schimel, Detecting drought impact on terrestrial biosphere carbon fluxes over contiguous US with satellite observations. *Environ Res Lett*. **13**, 095003 (2018).

B. Gaubert, B. B. Stephens, S. Basu, F. Chevallier, F. Deng, E. A. Kort, P. K. Patra, W. Peters, C. Rödenbeck, T. Saeki, D. Schimel, I. V. der Laan-Luijkx, S. Wofsy, Y. Yin, Global atmospheric CO2 inverse models converging on neutral tropical land exchange, but disagreeing on fossil fuel and atmospheric growth rate. *Biogeosciences*. **16**, 117–134 (2019).

S. Wolf, T. F. Keenan, J. B. Fisher, D. D. Baldocchi, A. R. Desai, A. D. Richardson, R. L. Scott, B. E. Law, M. E. Litvak, N. A. Brunsell, W. Peters, and I. T. van der Laan-Luijkx. Warm spring reduced carbon cycle impact of the 2012 US summer drought. Proceedings of the National Academy of Sciences, 113(21):5880–5885, 2016. doi: 10.1073/pnas.1519620113. URL http://www.pnas.org/content/113/21/5880.abstract.

***Original***: *Another potential limitation to the usefulness of the data is the underwhelming resolution. Monthly fluxes at 4 x 5 degree resolution are no longer really state-of-the-art. One of the arguments for using satellite measurements is the higher spatial resolution of the signals that can be resolved compared to the rather sparse surface-based network. This dataset is not exploiting to this strength.*

**Response**: We chose 4º x 5º to reflect the information content of the current available space-based $CO_2$ data, rather than an arbitrary grid scale, and we note this spatial resolution has already been scientifically successful (e.g., Liu et al., 2017; Bowman et al., 2017; Liu et al., 2018; Sellers et al., 2018). Before the launch of GOSAT and OCO-2, the tropics had been basically treated as a whole (e.g., Gurney et al., 2002; Baker et al., 2006; Schimel et al., 2015). The 4º x 5º resolution has both scientific value and manageable uncertainties. The estimated posterior flux uncertainty reflects the actual uncertainty as shown in the comparison to aircraft $CO_2$ observations (Figure 9 in the text). Publishing the dataset will make the dataset easily accessible for more specific regional studies and thus will facilitate rapid progress.

Liu, J., Bowman, K. W., Schimel, D. S., et al. (2017). *Contrasting carbon cycle responses of the tropical continents to the 2015–2016 El Nino. Science, 358 eaam5690.*

Sellers, P. J., D. S. Schimel, B. Moore, J. Liu, and A. Eldering, Observing Carbon Cycle-climate feedbacks from space, Proceedings of the National Academy of Sciences Jul 2018, 115 (31) 7860-7868; DOI: 10.1073/pnas.1716613115

J. Liu, K. Bowman, N. C. Parazoo, A. A. Bloom, D. Wunch, Z. Jiang, K. R. Gurney, D. Schimel, Detecting drought impact on terrestrial biosphere carbon fluxes over contiguous US with satellite observations. *Environ Res Lett*. **13**, 095003 (2018).

Gurney KR, Law RM, Denning AS *et al*. (2002) Towards robust regional estimates of $CO_2$ sources and sinks using atmospheric transport models. *Nature*, **415**, 626– 630.

Baker, D. F., et al. (2006), TransCom 3 inversion intercomparison: Impact of transport model errors on the interannual variability of regional CO$_2$ fluxes, 1988–2003, *Global Biogeochem. Cycles*, 20, GB1002, doi:10.1029/2004GB002439.

Schimel D, Stephens BB, Fisher JB. 2015. Effect of increasing CO$_2$ on the terrestrial carbon cycle. *Proceedings of the National Academy of Sciences, USA* **112**: 436– 441.

Bowman, K. W., Liu, J., Bloom, A. A., Parazoo, N. C., Lee, M., Jiang, Z., … Wunch, D. (2017). Global and Brazilian carbon response to El Niño Modoki 2011–2010. *Earth and Space Science*, 4, 637– 660. https://doi.org/10.1002/2016EA000204

***Original:*** *Other comments:*
*Regarding the completeness of the dataset presented, I had some minor concerns. I tried to check the availability of the datasets linked to here, and found that I was not sure which version of the ECCO-Darwin fluxes had been used: the data portal lists several different options. I was not even entirely sure if the ocean fluxes had been optimized, but the netCDF gridded fluxes describe the ocean fluxes as "posterior ocean fluxes,2010-2014 constrained by GOSAT, 2015-2018 constrained by OCO2", so I assume that they are not identical to the prior. In any case, this could be clarified.*
**Response**: In the revision, we included the prior ocean fluxes, prior biosphere fluxes, and fossil fuel emissions in the gridded product, so the dataset is complete and the readers can calculate carbon budget of any defined region.

***Original:*** *Similarly, the paper mentions that FLUXCOM-GPP is one of the inputs to CARDAMOM, but there is more than one version of this product as well. Even CARDAMOM comes in different flavours, I believe, based on the documentation in the cited papers. For completeness it would be suitable to include all the prior and posterior fluxes in the dataset - including the anthropogenic fluxes which are not optimised. Only then can the full budget be assessed.*
**Response**: In the revision, we specified the version of FLUXCOM GPP in the text: "In addition, mean GPP and fire carbon emissions from 2010 - 2017 are constrained by FLUXCOM RS+METEO version 1 GPP (Tramontana et al., 2016; Jung et al., 2017)".

In the revised dataset, we included all the prior and posterior fluxes in the dataset

***Original***: *I do not see the purpose of providing the monthly fluxes (with uncertainty) at 13 different FLUXNET sites. As far as I can tell, these are extracted directly from the model, and do not represent additional downscaling or enhanced temporal resolution. The benefit of this (and the rationale for the selection of these specific sites) is not clear to me. The measured monthly mean NBE at these sites is not included for comparison, nor is any validation using FLUXNET sites provided in the manuscript. It seems redundant.*
**Response**: We have removed the monthly fluxes at 13 different FLUXNET sites.

***Original:*** *I was surprised by the choice of the masks used for aggregation of fluxes. If two masks are included, why not include the broadly-applied TransCom mask? The benefit of such a common mask is the ease of comparison. Yes, a user may apply his or her own mask to the data, but it really does not add much in terms of space (22 regions with monthly resolution), and would facilitate comparison with already available model output. This is likely of more general application than the two custom masks given here.*
**Response**: In the revision, we added the monthly fluxes at TransCom regions (Figures 3, 5, and 8). We revised corresponding text to reflect the changes. In section 4.2, we added the following description: "The availability of flux estimates over the broadly used TransCom regions make it easy to compare to previous studies. For example, we estimate that the annual net carbon uptake over North America is 0.7 ± 0.1 GtC/year with 0.2 GtC variability between 2010 and 2018, which agrees with 0.7 ± 0.5 GtC/year estimates based on surface $CO_2$ observations between 1996-2007 (Peylin et al., 2013)." In addition, we revised section 4.4: "We provide the regional mean NBE seasonal cycle, its variability, and uncertainty based on the three regional masks (Table 5). Here we briefly describe the characteristics of the NBE seasonal cycle over the 11 TransCom regions, and its comparison to three independent top-down inversion results based on surface $CO_2$, which are CT-Europe (e.g., van der Laan-Luijkx et al., 2017) CAMS (Chevallier et al., 2005), and Jena CarbonScope (Rödenbeck et al., 2003). CMS-Flux-NBE differs the most from surface $CO_2$-based inversions over the South American Tropical, Northern Africa, tropical Asia, and NH boreal regions. The CMS-Flux NBE has a larger seasonal cycle amplitude over tropical Asia and Northern Africa, where the surface $CO_2$ constraint is weak, while it has a smaller seasonal cycle amplitude over the boreal region; this may be due to the sparse satellite observations over the high latitudes and weaker seasonal amplitude of the prior CARDAMOM fluxes. The comparison to FluxSat GPP can only qualitatively evaluate the NBE seasonal cycle, but cannot differentiate among different estimates."

***Original****: Minor/typographical comments:*
*L29: "from Greenhouse" -> "from the Greenhouse"*
*L30: remove "the" before NASA*
**Response:** We will correct the grammar**.**

**Original**: *L49: Crowell et al. 2019 is an odd choice as an example of inversions based on surface CO2 observations, as this was explicitly not the focus of the publication.*
**Response**: We replaced the reference with Chevallier et al., 2010.

***Original:*** *L55: The NBE are far -> NBE is far*
*L108: suggest " North America (NA)" -> "North American" (abbreviation is established elsewhere, and adjectival form fits better here)*
*L122: Section 8 is -> Section 8 describes the*
*L128: "that no" -> "no"*
*L151: its -> their*
*L161: from2010 (missing space)*
*L169: CARDAMON -> CARDAMOM*
*L184: by ACOS -> by the ACOS*
*L185: maximize -> maximizes*
*L193: land nadir good quality observations -> land nadir observations flagged as being of good quality*
*L221: of OCO-2 -> of the OCO-2*
*L272: over Pacific, and -> over the Pacific, but*
*L277: its -> their*
*L279: each nine -> each of nine*
*L283: fractions sampled at ith aircraft locations -> fraction sampled at the ith aircraft location*
*L285: of mean -> of the mean*
*L287: either posterior fluxes or transport -> either the posterior fluxes or the transport*

*L288: posterior fluxes -> the posterior fluxes (twice)*
*L315: of RMSE to posterior flux using GEOS-Chem -> of the RMSE to the posterior flux using the GEOS-Chem ăˇA*

*´l*
*L342: Please rewrite the first sentence.*
*L355 & L366: by NOAA -> by the NOAA*
*L360: of posterior -> of the posterior*
*L371: calculate -> calculated*
*L375: with GCP -> with the GCP (or, "with the range estimated by the GCP,")*
*L382: shows large -> shows that large*
*L382: Southern -> the Southern*
*L383: eastern -> the eastern*
*L409: or weakly -> or are weakly*
*L410: during 2015 -> during the 2015*
*L415: Pouter -> Poulter*
*L416: capitalisation weird in "tropical south America Savanna"ăˇA*
*´l*
*L440: above planetary*
*-> above the planetary*
*L446: used NOAA -> used the NOAA*
*L448: is equal or above -> is greater than or equal toăˇA*
*´l*
*L450 & L482 & L497: Southern Ocean -> the Southern Ocean (as an aside: the weaker seasonality certainly plays a role, but this was also a "problem region" in comparison to Atom-1 measurements, so perhaps there is something else going on there: : :)*

**Response**: In the revision, we have revised the text accordingly.

**Original**:: *Figure 2: It is really difficult to tell the regions apart on this map. The blues in North America, for instance, really blend together.*

**Response**: We remade Figure 2.

**Original:** *Figure 9: This figure seems unnecessarily cramped. Perhaps split it into two? For instance, I could barely tell that the bars were really blue without zooming way in, as they are so tiny.*

**Response**: In the revision, we removed the bars, since the number of observations has very little information. We believe that the new figure is clearer.

**Original**: *General figure comment: Something is a bit off with the rendering of the digits in your colour bars, making the bottom bar of "2"s disappear and making a gap in the bottom of round digits.*
**Response:** we have remade Figures 10 and 11.

**Original:{** *L466: from Monte -> from the Monte*
*L478: What is meant here? "either transport or low of posterior flux uncertainty estimates": Perhaps, "either transport errors or too low values for posterior flux uncertainties"?*
*L480: of flight -> of the flight*
*L493: defined -> as defined*
*L496: These -> The, also, specify which ratio you mean "between RMSE and RMSEMC"*
*L500: Pacific -> the Pacific*
*L503: these -> the*
*L504: with 4_x 5_resolution transport model -> using a transport model with only 4_x*

*5_resolution*

*L513-515: A bit awkward, please rephrase.*

*L516: indicates small -> indicates a small*

*L518: of posterior -> of the posterior*

*L533: to FLUXSAT -> to the FLUXSAT*

*L541: "needs caution" -> perhaps better: "calls for caution"*

*L556: by GCP -> by the GCP*

*L562: atmospheric -> the atmospheric*

*L563: level -> levels*

*L576: provide support the monitoring of the regional contributions to the changes inǍaatmospheric: : : I'm not sure about this, perhaps: "support the monitoring of the regional (biospheric) contributions to changes in atmospheric: : :"?*

*L580 & L582: data is -> data are*

*L582: ensemble posterior -> ensemble of posterior*

**Response**: We have incorporated all these comments in the revision.

Thanks for the constructive comments. Please see our response below.
Anonymous Referee #2

***Original***: *The authors present a brief description and evaluation results for the inverse model estimated CO2 fluxes for 2010-2018, based on observations by GOSAT and OCO-2 satellites. The data presented in the dataset are produced with the same model that was applied in several research papers and have mostly been used for estimating the variability and anomalies in the global carbon cycle at the regional and global scale. The satellite-based flux inversions proved to be useful in constraining large regional scale response of the natural carbon cycle to climate anomalies, droughts, heatwaves, such as those driven by the El-Nino cycle. In this context, the presented data can become a useful asset for those studying the carbon cycle variability at regional scale and its connection to the climate anomalies.*

*On the negative side, there are desirable components in the evaluation, such as analysis of the CO2 flux seasonal cycle, its comparison with inverse model estimates made with ground-based observations, or other independent estimates, such as based on flux tower data. The same can be said on comparison with observed CO2 concentration at background monitoring sites, such as the NOAA flask sampling network. In case there are identified biases in such comparison, it would be possible to advise the users to restrict the use of the data to studying the flux anomalies rather than using the fluxes for forward simulations, comparing with surface fluxes and using in ecosystem model optimization, where seasonal cycle performance is important. The authors should clearly state such limitations so that the users can have enough information on how to make best use of the provided data. The paper is well written and can be accepted after minor revision addressing the comments and suggestions.*

**Response**: We appreciate the constructive comments. In the revision, we added the comparison to NOAA marine boundary layer reference sites, and compared the seasonal cycle to three publicly available surface CO2 constrained top-down flux inversions. Please see our detailed response below.

Detailed comments.
***Original***: *Notable deficiency: NBE flux evaluation looks somewhat qualitative. Based on data presented in the paper, and data provided on the data distribution site it is difficult to compare the NBE fluxes to alternative estimates. The 28-region data is provided, but it doesn't look directly mappable to widely used Transcom-3 22 region map. Recommend adding comparison figure (similar to Figure 8) of the seasonal flux climatology on Transcom3 22 regions or the authors-proposed 28 regions to other available estimates such as CAMS inversion fluxes (based on Chevallier et al. 2010) or FLUXCOM fluxes (Jung et al. 2020).*

**Response**: In the revision, we added monthly fluxes at TransCom 3 regions as part of the dataset in addition to the monthly fluxes at the original two regional masks we used in the first version. We revised Figure 3 to include the TransCom 3 region mask, and Figure 5 to include climatological NBE fluxes, variability, and uncertainties at TransCom 3 regions.

We replaced Figure 8 with NBE seasonal cycle comparisons between CMS-Flux NBE and the three publicly available surface $CO_2$ constrained top-down NBE estimates, which are CAMS, Jena CarbonScope, and CT-Europe. We revised section 4.4 based on the new Figure 8: "We provide the regional mean NBE seasonal cycle, its variability, and uncertainty based on the three regional masks (Table 5). Here we briefly describe the characteristics of the NBE seasonal cycle over the 11 TransCom regions, and its comparison to three independent top-down inversion results based on surface $CO_2$, which are CT-Europe (e.g., van der Laan-Luijkx et al., 2017) CAMS (Chevallier et al., 2005), and Jena CarbonScope (Rödenbeck et al., 2003). CMS-Flux-NBE differs the most from surface $CO_2$-based inversions over the South American Tropical, Northern Africa, tropical Asia, and NH boreal regions. The CMS-Flux NBE has a larger seasonal cycle amplitude over tropical Asia and Northern

Africa, where the surface $CO_2$ constraint is weak, while it has a smaller seasonal cycle amplitude over the boreal region; this may be due to the sparse satellite observations over the high latitudes and weaker seasonal amplitude of the prior CARDAMOM fluxes. The comparison to FluxSat GPP can only qualitatively evaluate the NBE seasonal cycle, but cannot differentiate among different estimates. In general, the months that have larger productivity corresponds to months with a net uptake of carbon from the atmosphere, especially over the NH (Figure 8). More research is still needed to understand the seasonal cycles of NBE, including its phase (i.e., transition from source to sink) and amplitude (peak-to-trough difference), and its relationships between GPP and respiration."

*Original*: Line 208 It looks like presented bias figures (below 0.1 ppm) are related to global mean bias, are the bias values available as seasonal mean values by latitude or TCCON site?
Are retrieved and bias-corrected concentrations consistent with model simulations optimized with ground-based observations?
**Response:** O'Dell et al. (2018) compared OCO-2 $X_{CO2}$ with observations from TCCON sites and the model simulations optimized with ground-based CO2. In the revision, we cited O'Dell et al. (2018), and added the following discussion in section 2.3: "O'Dell et al. (2018) showed that the OCO-2 $X_{CO2}$ land nadir retrievals has a mean bias ~0.3 ppm and RMS ~1.1 ppm when compared to TCCON retrievals; the differences between OCO-2 $X_{CO2}$ retrievals and surface $CO_2$ constrained model simulations are well within 1.0 ppm over most of the locations in the Northern Hemisphere (NH), where most of the surface $CO_2$ observations are located."

O'Dell, C. W., Eldering, A., Wennberg, P. O., Crisp, D., Gunson, M. R., Fisher, B., Frankenberg, C., Kiel, M., Lindqvist, H., Mandrake, L., Merrelli, A., Natraj, V., Nelson, R. R., Osterman, G. B., Payne, V. H., Taylor, T. E., Wunch, D., Drouin, B. J., Oyafuso, F., Chang, A., McDuffie, J., Smyth, M., Baker, D. F., Basu, S., Chevallier, F., Crowell, S. M. R., Feng, L., Palmer, P. I., Dubey, M., García, O. E., Griffith, D. W. T., Hase, F., Iraci, L. T., Kivi, R., Morino, I., Notholt, J., Ohyama, H., Petri, C., Roehl, C. M., Sha, M. K., Strong, K., Sussmann, R., Te, Y., Uchino, O., and Velazco, V. A.: Improved retrievals of carbon dioxide from Orbiting Carbon Observatory-2 with the version 8 ACOS algorithm, Atmos. Meas. Tech., 11, 6539–6576, https://doi.org/10.5194/amt-11-6539-2018, 2018.

*Original:* Line 1031 Figure 8. Although the seasonally varying fluxes look to be in a reasonable range it is very much advisable to compare/plot along with observed or observation-based fluxes, such as FLUXCOM NEE product (Jung et al. 2020).

**Response:** See our response above.

Technical corrections.
*Original*: Line 191 Looks anomalous, to have 2000 good quality retrievals available on a single day in the ACOS-GOSAT dataset (appears significantly larger than average).
**Response**: The number 2000 is the number of soundings that are processed by retrieval algorithm, not the number of good quality observations. The number of good quality retrievals is between ~100-300 daily.

*Original:* Line 193 Need to state how good quality is defined (what value of the quality flag is used)?
**Response**: We clarified in the revision: "We only assimilate ACOS-GOSAT land nadir good quality observations, which are the retrievals with quality flag equal to 1."

*Original*: Line 199 Is 'super observations' a good term to name 100 km (_12 sec) average data?

**Response**: This term was originated from numerical weather prediction. In the revision, we cited a relevant reference to further clarify the term: "To reduce the sampling error due to the resolution differences between the transport model and OCO-2 observations, we generate super observations by aggregating the observations within ~100 km (along the same orbit) (Liu et al., 2017). The super-obbing strategy was first proposed in numerical weather prediction (NWP) to assimilate dense observations (Lorenc, 1981), and is still broadly used in NWP (e.g., Liu and Rabier, 2003)."

*Original: Line 228 The statement "For large-order systems, the posterior errors cannot be explicitly calculated" can be argued. Posterior flux uncertainty projected to regions can be estimated analytically using recipes provided by (Fisher and Courtier, 1995) or (Meirink et al, 2008), using either flux singular vectors or flux increments obtained on course of the iterative optimization (eg Niwa and Fujii, 2020). Using random perturbations is simpler and is used widely, but that doesn't mean that the more accurate method is impossible to apply.*

**Response**: In section 2.4, we incorporated the above comments when we discuss the posterior flux uncertainty estimation: "Posterior flux uncertainty projected to regions can be estimated analytically based on the methods described in Fisher and Courtier (1995) or Meirink et al. (2008), using either flux singular vectors or flux increments obtained during the iterative optimization (e.g., Niwa and Fujii, 2020). In this study, we rely on a Monte Carlo approach to quantify posterior flux uncertainties following Chevallier et al. (2010) and Liu et al. (2014), which is simpler and widely used."

*Original: Line 240 Common perception is that tower footprint size is less than 1 km, based on estimates by Baldocchi, (1997) and others. The citation by Running et al (1999) of 'several km2' may refer to the upper range. They (Running et al 1999) also consider 1–3 km2 and 1 km2 as typical values throughout their paper.*

**Response**: We revised the description to: "Direct NBE estimates from flux towers only provide a spatial representation of roughly 1 – 3 kilometers (Running et al., 1999),…".

[revised manuscript text omitted]

---

## Author Response (AR3)

We appreciate very much of the comments. In the following, the original comments are marked
green followed by our reply with no color. Following our response is the full paper with tracked
changes.

Comments to the Author:
Page 19, lines 493-494: Author cite Friedlingstein 2019 for the most recent GCB but their
summary in these lines misses two source terms of the global carbon budget. Technically I think
we would call annual growth in atmospheric CO2 concentrations the 'net' of sources and sinks,
not the 'sum'.
In the revision, we revised the sentence to "The annual atmospheric $CO_2$ growth rate, which is
the net difference between fossil fuel emissions and total annual sink over land and ocean…"
Page 20, lines 505-506: Authors change units here (from GtC to %) but they have not quoted
GCB correctly. For decade 2009-2018, GCB has ocean sink at 2.5 +/- 0.6 GtC yr-1 and land sink
at 3.2 +/- 0.6 GtC yr-1. Land significantly greater than ocean in GCB while these authors have
ocean greater than land. Something wrong somewhere. Reader can not accept (line 507) that
these numbers fit within uncertainty of GCB numbers.
This comment arises from the misunderstanding of the NBE definition. The land sink at $3.2 \pm 0.6$
GtC from GCB does not include land use changes and residual imbalance, while the NBE we
report here includes all land fluxes except fossil fuel emissions. We define the term "NBE" in the
introduction as: "The net biosphere exchange (NBE), which is the net carbon flux of all the land-
atmosphere exchange processes except fossil fuel emissions …".
In section 4.1, we further clarify how we calculate NBE from GCB – 2019 reported fluxes: "The
GCB does not report NBE directly, we calculate NBE from GCB-2019 as the residual differences
between fossil fuel, ocean net carbon sink, and atmospheric $CO_2$ growth rate. It is also equivalent
to ($S_{LAND} + B_{IM} - E_{LUC}$) reported by Freidlingstein et al., 2019, where $S_{LAND}$ is terrestrial sink, $B_{IM}$
is a budget imbalance, and $E_{LUC}$ is land use change."
Authors here refer to GCP (Global Carbon Project?)? Friedlingstein 2019 refers to GCB: global
carbon budget. Neither Friedlingstein 2019 nor Friedlingstein 2020 make any mention of NBE.
This section (4.1) needs complete rewrite: use correct terms, adopt consistent units (GtC or %) but
not both, line 507 conflicts with line 516-517. NBE as defined here does not match terms from
GCB. What happened to cement? Do these authors adopt the same +/- 1 sigma as GCB? ESSD
readers who might take an interest in this product will very likely know GCB in good detail to
detect deficiencies here.
See our response above. Even though Friedlingstein et al. (2019, 2020) do not report the term NBE,
they calculate ($S_{LAND} + B_{IM} - E_{LUC}$) when comparing to fluxes from top-down atmospheric flux
inversions (Figure 8 in Friedlingstein et al. ,2019), which is equivalent to NBE we report here.
In the revision, we have added the fluxes in GtC in section 4.1.  Cement is counted in the fossil
fuel emissions.
In section 2.4, we detailed the uncertainty quantification method.

Line 552 - TER not defined. Not defined until line 712.
TER is defined in the first sentence of section 2.5.1.
Line 733: Authors refer to GCP when they mean GCB. Fix all of these errors! GCP 2019 not a valid
citation, Friedlingstein 2019 is.
In the revision, we have removed the GCP and replaced it with GCB-2019.
Line 738 to 746, illogical. Continental US (CONUS) and state-level of CONUS have very good
observational data (ground and aircraft). Therefore, observational basis of CO2 emission for these
particular regions (authors do NOT go to state level) will likely always prove more reliable (less
uncertain) than satellite xCO2. Unreliable estimates of NBE, no matter how well distributed over
CONUS, will never prove more reliable than obs here. But validation here can add credibility to xCO2
inversions elsewhere? Authors current paragraph implies that because NBE varies a lot regionally or
seasonally, it will prove useful here. Not likely. 1.5 C target (hopeful) comes from UNFCCC CoP21
Paris Agreement, not from IPCC AR6. IPCC AR6 never cited here.
What we want to convey in that paragraph is that it is important to monitor the changes of NBE
since the regional contributions to atmospheric CO2 growth rate is the sum of NBE and fossil fuel
emissions. It seems that the US example generates confusion. In the revision, we deleted the
sentence: "Even over the continental US, where fossil fuel emissions are ~1.5 GtC/year, the
changes of regional NBE in the future can significantly modify regional contributions to the
changes of atmospheric $CO_2$ (Liu et al., 2018)."
The 1.5 C target is discussed in IPCC special report, 2018. We revised the citation in the revision.
Supplement consists entirely of figures, with some redundancy to main text. Because Copernicus does
not archive supplements, this supplement needs to go on the JPL site or these 11 figures need to go in
the Appendix.
I am perplexed here, since I do see ESSD papers with supplement materials published online. Here
are a few examples:
Yu, Q., You, L., Wood-Sichra, U., Ru, Y., Joglekar, A. K. B., Fritz, S., Xiong, W., Lu, M., Wu,
W., and Yang, P.: A cultivated planet in 2010 – Part 2: The global gridded agricultural-production
maps, Earth Syst. Sci. Data, 12, 3545–3572, https://doi.org/10.5194/essd-12-3545-2020, 2020.
McDuffie, E. E., Smith, S. J., O'Rourke, P., Tibrewal, K., Venkataraman, C., Marais, E. A., Zheng,
B., Crippa, M., Brauer, M., and Martin, R. V.: A global anthropogenic emission inventory of
atmospheric pollutants from sector- and fuel-specific sources (1970–2017): an application of the
Community Emissions Data System (CEDS), Earth Syst. Sci. Data, 12, 3413–3442,
https://doi.org/10.5194/essd-12-3413-2020, 2020.
If it is still possible, we prefer to have supplement instead of having figures in the Appendix. But
to accelerate the publication of the paper, in the revision, we have put the supplement figures in
the Appendix B.

[revised manuscript text omitted]